



# Reduced El Niño variability in the mid-Pliocene according to the PlioMIP2 ensemble

Arthur M. Oldeman[1], Michiel L. J. Baatsen[1], Anna S. von der Heydt[1,2], Henk A. Dijkstra[1,2], Julia C. Tindall[3], Ayako Abe-Ouchi[4], Alice R. Booth[5], Esther C. Brady[6], Wing-Le Chan[4], Deepak Chandan[7], Mark A. Chandler[8], Camille Contoux[9], Ran Feng[10], Chuncheng Guo[11], Alan M. Haywood[3], Stephen J. Hunter[3], Youichi Kamae[12], Qiang Li[13], Xiangyu Li[14], Gerrit Lohmann[15], Daniel J. Lunt[16], Kerim H. Nisancioglu[11], Bette L. Otto-Bliesner[6], W. Richard Peltier[7], Gabriel M. Pontes[17], Gilles Ramstein[9], Linda E. Sohl[8], Christian Stepanek[15], Ning Tan[9,18], Qiong Zhang[13], Zhongshi Zhang[14], Ilana Wainer[17], and Charles J. R. Williams[16,19]

[1]Institute for Marine and Atmospheric research Utrecht (IMAU), Department of Physics, Utrecht University, Utrecht, The Netherlands.
[2]Centre for Complex Systems Science, Utrecht University, Utrecht, The Netherlands
[3]School of Earth and Environment, University of Leeds, Woodhouse Lane, Leeds, West Yorkshire, LS29JT, UK
[4]Atmosphere and Ocean Research Institute, The University of Tokyo, Kashiwa, 277-8564, Japan
[5]School of Ocean and Earth Science, University of Southampton, National Oceanography Centre, Southampton, UK
[6]National Center for Atmospheric Research, (NCAR), Boulder, CO 80305, USA
[7]Department of Physics, University of Toronto, Toronto, M5S 1A7, Canada
[8]CCSR/GISS, Columbia University, New York, NY 10025, USA
[9]Laboratoire des Sciences du Climat et de l'Environnement, LSCE/IPSL, CEA-CNRS-UVSQ Université Paris-Saclay, 91191 Gif-sur-Yvette, France
[10]Department of Geosciences, College of Liberal Arts and Sciences, University of Connecticut, Storrs, CT 06033, USA
[11]NORCE Norwegian Research Centre, Bjerknes Centre for Climate Research, 5007 Bergen, Norway
[12]Faculty of Life and Environmental Sciences, University of Tsukuba, Tsukuba, 305-8572, Japan
[13]Department of Physical Geography and Bolin Centre for Climate Research, Stockholm University, Stockholm, 10691, Sweden
[14]Department of Atmospheric Science, School of Environmental studies, China University of Geoscience, Wuhan 430074, China
[15]Alfred-Wegener-Institut – Helmholtz-Zentrum für Polar and Meeresforschung (AWI), Bremerhaven, 27570, Germany
[16]School of Geographical Sciences, University of Bristol, Bristol, BS8 1QU, UK
[17]Oceanographic Institute, University of São Paulo, Brazil
[18]Key Laboratory of Cenozoic Geology and Environment, Institute of Geology and Geophysics, Chinese Academy of Sciences, Beijing 100029, China
[19]NCAS-Climate, Department of Meteorology, University of Reading, Reading, UK

**Correspondence:** Arthur M. Oldeman (a.m.oldeman@uu.nl)

**Abstract.** The mid-Pliocene warm period (3.264 – 3.025 Ma) is the most recent geological period during which atmospheric $CO_2$ levels were similar to recent historical values (∼400 ppm). Several proxy reconstructions for the mid-Pliocene show highly reduced zonal sea surface temperature (SST) gradients in the tropical Pacific Ocean, indicating an El Niño-like mean state. However, past modelling studies do not show these highly reduced gradients. Efforts to understand mid-Pliocene climate
dynamics have led to the Pliocene Model Intercomparison Project (PlioMIP). Results from the first phase (PlioMIP1) showed





clear El Niño variability (albeit significantly reduced) and did not show the greatly reduced time-mean zonal SST gradient suggested by some of the proxies.

In this work, we study El Niño-Southern Oscillation (ENSO) variability in the PlioMIP2 ensemble, which consists of additional global coupled climate models and updated boundary conditions compared to PlioMIP1. We quantify ENSO amplitude,

period, spatial structure and flavour, as well as the tropical Pacific annual mean state in mid-Pliocene and pre-industrial simulations. Results show a reduced ENSO amplitude in the model-ensemble mean (-24%) with respect to the pre-industrial, with 15 out of 17 individual models showing such a reduction. Furthermore, the spectral power of this variability considerably decreases in the 3 – 4 year band. The spatial structure of the dominant empirical orthogonal function shows no particular change in the patterns of tropical Pacific variability in the model-ensemble mean, compared to the pre-industrial. Although

the time-mean zonal SST gradient in the equatorial Pacific decreases for 14 out of 17 models (0.2 °C reduction in the ensemble mean), there does not seem to be a correlation with the decrease in ENSO amplitude. The models showing the most 'El Niño-like' mean state changes show a similar ENSO amplitude as in the pre-industrial reference, while models showing more 'La Niña-like' mean state changes generally show a large reduction in ENSO variability. The PlioMIP2 results show a reasonable agreement both with time-mean proxies indicating a reduced zonal SST gradient, as well as reconstructions indicating a

reduced, or similar, ENSO variability.

## 1 Introduction

The mid-Piacenzian or mid-Pliocene warm period (mPWP, 3.264 - 3.025 Ma) was a recent geological interval of sustained warmth with global mean temperatures 2 - 5 °C higher than pre-industrial (Haywood et al., 2010; Dowsett et al., 2010, 2016; Haywood et al., 2020). Atmospheric $CO_2$ levels were around $\sim$ 400 ppm (Badger et al., 2013; Fedorov et al., 2013), similar

to recent historical values, although recent reconstructions suggest it was closer to $\sim$ 370 ppm (de la Vega et al., 2020). This makes this period an interesting case study for our near-future climate, also because the mid-Pliocene had a similar geography to the present (outside of ice-sheet regions). Efforts to understand the mPWP climate have been ongoing for more than 25 years, and led to the coordination of the Pliocene Modelling Intercomparison Project (PlioMIP) phase 1 in 2010 (Haywood et al., 2010). The PlioMIP1 ensemble shows a range of global mean surface temperature anomalies, even though the models have

nearly identical boundary conditions. Furthermore, comparison with proxies highlights that most models underestimate polar amplification (Haywood et al., 2013). The PlioMIP phase 2 was initiated to further understand the mPWP climate, and more specifically designed to reduce uncertainties in model boundary conditions and in proxy data reconstruction (Haywood et al., 2016a, 2020). It employs boundary conditions from the Pliocene Research, Interpretation and Synoptic Mapping (PRISM) version 4, including updated reconstructions of ocean bathymetry and land-ice surface topography, as well as Pliocene soils and

lakes (Dowsett et al., 2016; Haywood et al., 2016a). Important changes in boundary conditions compared to the experimental design of PlioMIP1 include the closure of the Canadian Archipelago and the Bering Strait, and the shoaling of the Sahul and Sunda shelves. The ensemble mean global annual mean surface air temperature (SAT) increases with 3.2 °C (1.7 - 5.2 °C) compared to the pre-industrial, when implementing PRISM4 boundary conditions in PlioMIP2 (Haywood et al., 2020).



One of the more perplexing and still unanswered topics in the Pliocene research community is the behaviour of tropical
Pacific variability in the mid-Pliocene, in particular of the El Niño-Southern Oscillation (ENSO). In the present-day climate,
ENSO is the most prominent mode of variability on interannual time scales. It has its origin in the tropical Pacific, while having
teleconnections to many regions in the world (Philander, 1990). The ENSO phenomenon can be explained as an internally
generated mode of variability of the coupled equatorial ocean-atmosphere system - either self-sustained or excited by random
noise (Fedorov et al., 2003). The background climate such as meridional and zonal sea surface temperature (SST) gradients,
vertical temperature gradients as well as the trade wind strength is thought to play an important role in the properties of this
internal mode of variability. In future projections it is still unclear how ENSO behaviour will be affected by global warming
(Kim et al., 2014; Brown et al., 2020). It is therefore an interesting issue to study ENSO variability in the mid-Pliocene.

Early proxy reconstructions indicated that the mid-Pliocene tropical Pacific may have highly reduced zonal SST gradients
(Molnar and Cane, 2002; Wara et al., 2005; Ravelo et al., 2006). This pointed in the direction of an 'El Niño-like' mean state
in the mid-Pliocene, and it was even suggested that there might be a 'permanent El Niño' without any interannual variability
around that state (Fedorov et al., 2006). The coarse temporal resolution and choice of calibration of ocean sediment proxy
reconstructions make it challenging to say anything about such variability. Because of this, even a 'La Niña-like' mean state
has been proposed for the mPWP (Rickaby and Halloran, 2005). Scroxton et al. (2011) present evidence for clear ENSO
variability in the Pliocene based on ocean sediment isotopes, possibly despite reduced zonal SST gradients. Watanabe et al.
(2011) present coral skeleton data showing ENSO variability that is similar to the present-day. Mg/Ca measurements indicating
subsurface temperatures in the western equatorial Pacific again point towards a reduced zonal temperature gradient (Ford et al.,
2015), while more recent alkenone proxy reconstructions show a moderate reduction in the tropical Pacific zonal SST gradient
(Tierney et al., 2019). Latest proxy reconstructions by White and Ravelo (2020) show that the mid-Pliocene ENSO amplitude
varied between reduced and similar to present-day, and they associate that to a weaker thermocline feedback.

The earliest modelling studies on the early- and mid-Pliocene ENSO do not agree, with some studies showing little variability
(Fedorov et al., 2006; Barreiro et al., 2006) and others clearly showing ENSO-like interannual variability (Haywood et al., 2007;
Bonham et al., 2009; von der Heydt et al., 2011). While most modelling studies show (slightly) reduced zonal SST gradients,
a study using the Zebiak-Cane model of the tropical Pacific suggests a westward shift in the position of the cold-tongue under
increased background temperatures, whereas a weaker zonal SST gradient would be mostly associated with weaker background
trade winds (von der Heydt et al., 2011). Later, in the coordinated modelling efforts of PlioMIP1, all models show ENSO-like
variability, and 8 out of 9 studies agree on a reduced ENSO variability compared to the pre-industrial (Brierley, 2015), although
the magnitude of reduction varies considerably among different models. This robustly weaker ENSO is accompanied with a
shift to lower frequencies (or longer periods) in most models, while again the magnitude of the dominant frequency varies
among the model ensemble. Moreover, the PlioMIP1 models do not show a consistent reduction in the mean zonal SST
gradient, and a clear reason for weaker ENSO variability is not found. Research with the HadCM3 model has pointed out the
importance of centennial scale variability in ENSO behaviour, suggesting there could have existed periods with both weaker
and stronger ENSO variability in the mid-Pliocene (Tindall et al., 2016).





Modelling efforts on past as well as future climates show different responses of ENSO variability to radiative and geographical forcings. Collins et al. (2010) show that from the CMIP3 ensemble, it is not possible to determine whether the amplitude and frequency of ENSO variability will change in the future under climate change. Also the more recent CMIP5 ensemble provides no clear consensus on whether ENSO amplitude would decrease or increase in the future, and what feedbacks might change (Kim et al., 2014). However, Cai et al. (2014) do suggest a shift to higher ENSO frequencies under global warming based on the CMIP5 ensemble. Yeh et al. (2009) and Ashok and Yamagata (2009) show that the 'flavour' of El Niño will change in the future, shifting from mainly cold tongue (CT) El Niño events to more warm pool (WP) El Niño events, implying that the largest temperature variations will shift more towards the central Pacific. When reproducing the climate in recent decades, climate models suggest a reduced zonal SST gradient in the tropical Pacific due to rising greenhouse gas concentrations, while observations show a strengthened gradient (Coats and Karnauskas, 2017). This discrepancy between coupled models and observations is attributed to the cold bias in the equatorial cold tongue by Seager et al. (2019). However, Heede et al. (2020) show that the initial transient response to $CO_2$ forcing is characterized by a strengthened zonal SST gradient, while the equilibrium response shows a warmer cold tongue. Brown et al. (2020) investigate ENSO both in the mid-Holocene, last glacial maximum (*lgm*) and last interglacial (*lig*) simulations of PMIP3/4 and future scenarios of CMIP5/6. They find a clear decrease in ENSO variability in the *lig* and mid-Holocene simulations, despite a stronger zonal SST gradient. Closer inspection demonstrates no clear correlation between the mean zonal SST gradient in the tropical Pacific and ENSO amplitude, when considering the PMIP3/4 and CMIP5/6 ensembles.

While the PlioMIP1 ensemble was able to adequately reproduce many of the spatial patterns in surface temperature as reconstructed from proxies, a number of uncertainties and model-data mismatches remained, in particular regarding the warming in high latitudes (Haywood et al., 2013). High-latitude temperatures clearly affect also the tropical climate, for example through effects on the Hadley Cell and trade-wind strength caused by an altered equator-to-pole temperature gradient. In order to reduce uncertainties and model-data mismatches, a completely new reconstruction of palaeogeography was used in PlioMIP2 as boundary condition for the models. Analysis of large-scale features of the ensemble by Haywood et al. (2020) shows a larger SAT anomaly compared to PlioMIP1, because of the inclusion of more models with a higher equilibrium climate sensitivity (ECS). It is also shown that the ensemble mean SSTs agree well with newly reconstructed SST proxies by Foley and Dowsett (2019). The PlioMIP2 ensemble also agrees well with reconstructions from a recent SST synthesis study focusing on the same mid-Pliocene time-slice (McClymont et al., 2020).

In this work, we study changes in ENSO variability in the PlioMIP2 ensemble compared to the pre-industrial and relate this to differences in the mean background climate of the tropical Pacific. In Section 2, we briefly introduce the models that participate in the PlioMIP2 ensemble and describe the methods used to analyse ENSO. Following this, we investigate ENSO variability in PlioMIP2 in terms of amplitude, frequency, spatial structure and 'flavour', and compare this to pre-industrial reference simulations in Section 3.1. Next, we investigate the relation between ENSO amplitude and the mean zonal SST gradient and study whether the tropical Pacific mean state is 'El Niño-like' in the mid-Pliocene, in Section 3.2. In Section 4, the results will be discussed in light of observations as well as intermodel differences within the ensemble. We will conclude with a summary and outlook.



## 2 Methods

### 2.1 The PlioMIP2 ensemble

Seventeen climate models form the PlioMIP2 ensemble, which is almost double the size of the PlioMIP1 ensemble. A list of the models, performing institute and reference to the work describing the individual models in more detail, is presented in Table 1. All models have performed simulations following the PlioMIP2 experimental protocol (Haywood et al., 2016b), providing both the pre-industrial control experiment ($E^{280}$) and the mid-Pliocene experiment ($Eoi^{400}$) data. Pre-industrial simulations are forced with ~280 ppmv atmospheric $CO_2$ concentrations. The mid-Pliocene simulations are forced with 400 ppmv $CO_2$ and all models apart from HadGEM3 use significantly different geographic boundary conditions, including closed Arctic Ocean gateways (Bering Strait, Canadian archipelago) and reduced land ice coverage (Greenland ice sheet, West Antarctic ice sheet). Results of several large-scale features, such as global mean surface air temperature (SAT), polar amplification factor and equilibrium climate sensitivity (ECS) of the PlioMIP2 ensemble, are presented in Haywood et al. (2020). Note that the results of the HadGEM3 model were not included in that paper, as the simulations finished after time of writing. Details for HadGEM3 can be found in Williams et al. (2021). More details on the $Eoi^{400}$ simulations of the individual models can be found in the references listed in Table 1.

Each modelling group has provided (at least) 100 years of both the $E^{280}$ and $Eoi^{400}$ simulation for analysis. In this work, we consider the last 100 years of monthly SST data in order to quantify and investigate ENSO variability. Data were regridded onto a regular $1° \times 1°$ grid using a bilinear interpolation, in order to analyse the model results in the same way. In the Supplement, we discuss the robustness of our analysis methods using the results of two ensemble members, using 500 years of data.

### 2.2 Analysis methods

#### 2.2.1 Niño indices

The Niño3.4 index, defined as the monthly SST anomaly in the Niño3.4 region in the equatorial Pacific, is most commonly used in present-day ENSO analysis. It can be used to determine the amplitude and period of ENSO variability. The Niño3.4 region is used since it shows the largest correlation with SST variability in the whole tropical Pacific. However, the question remains as to whether this is also true for the mid-Pliocene ENSO.

Figure 1 shows the standard deviation (s.d.) of the SST anomalies in the tropical Pacific for (a) the pre-industrial $E^{280}$ ensemble mean and (b) the mid-Pliocene $Eoi^{400}$ ensemble mean. Indicated in the plot are the four commonly used regions to study ENSO variability: the Niño4, Niño3.4, Niño3 and Niño1+2 regions. The magnitude of SST variability decreases in the mid-Pliocene equatorial Pacific, but it seems that the region with the largest ENSO-related variability keeps its position.

To determine the SST anomaly pattern with largest variance, we perform principal component analysis (PCA) on the monthly SST anomalies. As the climatology is substracted, we expect the leading principal component (PC1) in the tropical Pacific to capture ENSO variability. We correlate the PC1 with the four different Niño indices to check which region is representative for



**Table 1.** Details on the models contributing to the PlioMIP2 ensemble. More details (e.g. on treatment of sea-ice and vegetation) can be found in Haywood et al. (2020) and Williams et al. (2021).

| Model | Institute, country | Atmosphere resolution | Ocean resolution | CMIP?** | Eoi$^{400}$ reference |
|---|---|---|---|---|---|
| CCSM4 (CESM1.0.5) | NCAR, USA | FV0.9x1.25 ($\sim 1°$) (CAM4) | G16 ($\sim 1°$) | CMIP5 | (Feng et al., 2020) |
| CCSM4-UoT | UoT, Canada | as CCSM4 | as CCSM4 * | - | (Peltier and Vettoretti, 2014; Chandan and Peltier, 2017, 2018) |
| CCSM4-Utrecht (CESM1.0.5) | IMAU, the Netherlands | FV $2.5° \times 1.9°$ | as CCSM4 * | - | (Baatsen et al., 2021, in prep.) |
| CESM1.2 | NCAR, USA | FV0.9x1.25 ($\sim 1°$) (CAM5) | G16 ($\sim 1°$) | - | (Feng et al., 2020) |
| CESM2 | NCAR, USA | FV0.9x1.25 ($\sim 1°$) (CAM6) | G17 ($\sim 1°$) | CMIP6 | (Feng et al., 2020) |
| COSMOS | AWI, Germany | T31 ($3.75° \times 3.75°$) | GR30 ($3.0° \times 1.8°$) | - | (Stepanek et al., 2020) |
| EC-Earth 3.3 | Stockholm University, Sweden | $1.125° \times 1.125°$ | $1.0° \times 1.0°$ | CMIP6 | (Zheng et al., 2019) |
| GISS2.1G | GISS, USA | $2.0° \times 2.5°$ | $1.0° \times 1.25°$ | CMIP6 | - |
| HadCM3 | University of Leeds, UK | $2.5° \times 3.75°$ | $1.25° \times 1.25°$ | CMIP5 | (Hunter et al., 2019) |
| HadGEM3-GC31-LL | University of Bristol, UK | N96 ($1.875°$ x $1.25°$) | $\sim 1.0°$ x $1.0°$ | CMIP6 | (Williams et al., 2021) |
| IPSLCM5A | LSCE, France | $3.75° \times 1.9°$ | $0.5 - 2.0° \times 2.0°$ | CMIP5 | (Tan et al., 2020) |
| IPSLCM5A2.1 | LSCE, France | as IPSLCM5A * | as IPSLCM5A * | - | (Tan et al., 2020) |
| IPSLCM6A-LR | LSCE, France | $2.5° \times 1.1.26°$ | $1.0° \times 1.0°$, refined at $1/3°$ in the tropics | CMIP6 | (Lurton et al., 2020) |
| MIROC4m | JAMSTEC, Japan | T42 ($\sim 2.8° \times 2.8°$) | $0.5° - 1.4° \times 1.4°$ | - | (Chan and Abe-Ouchi, 2020) |
| MRI-CGCM 2.3 | MRI, Japan | T42 ($\sim 2.8° \times 2.8°$) | $0.5° - 2.0° \times 2.5°$ | CMIP5 | (Kamae et al., 2016) |
| NorESM-L | BCCR, Norway | T31 ($\sim 3.75° \times 3.75°$) (CAM4) | G37 ($\sim 3.0° \times 3.0°$) | - | (Li et al., 2020) |
| NorESM1-F | BCCR, Norway | FV19 ($1.9° \times 2.5°$) (CAM4) | $\sim 1.0° \times 1.0°$ | CMIP6 | (Li et al., 2020) |

* Slight differences, check Haywood et al. (2020) for details.

** Models that contributed to the Coupled Model Intercomparison Project (CMIP) phase 5 or 6.

ENSO variability, both in the pre-industrial simulations as well as in the mid-Pliocene. The results per model are presented in
Table 2.

    In the pre-industrial simulations, 14 of the 17 models show the largest correlation in the Niño3.4 region. The ensemble mean shows the largest correlation for the Niño3.4 index and the results agree well with data obtained from 1920-2020 HadISST





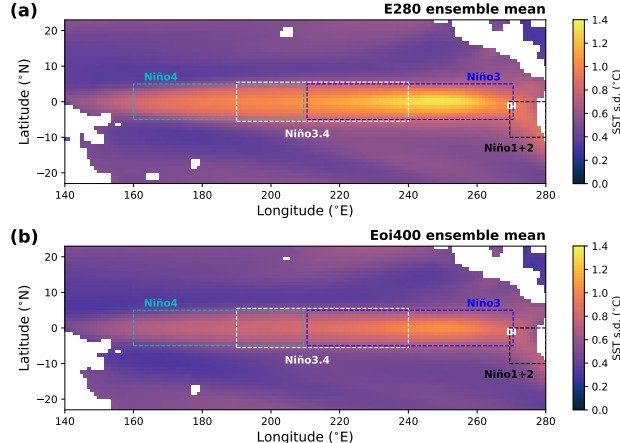

**Figure 1.** The ensemble mean standard deviation (s.d.) of SST anomalies in the tropical Pacific. **(a)** Results for pre-industrial E[280] and **(b)** mid-Pliocene Eoi[400] simulations.

observations (Rayner et al., 2003). Of the mid-Pliocene simulations, 15 of the 17 models show the largest correlation with the Niño3.4 index. The mid-Pliocene ensemble mean clearly shows the largest correlation in the Niño3.4 region. Henceforth, we

will be using the Niño3.4 index to quantify ENSO variability. Furthermore, some of the analyses performed in this study have also been repeated with the Niño3 index instead of the Niño3.4 index, resulting in the same conclusions.

### 2.2.2    Quantifying ENSO variability

In order to quantify ENSO variability within the PlioMIP2 ensemble, we look at four main features of ENSO: 1. Amplitude, 2. Period, 3. Spatial structure and 4. El Niño flavour (see below for explanation and specific definitions used). To compute

the amplitude and period we use the Niño3.4 index. Anomalies are taken with respect to a mean seasonal cycle computed on the full 100 year data that is available. No running mean is applied. Before any analysis, linear trends are removed from the Niño3.4 time series. The analysis methods used here are similar to those in the analysis of ENSO in the PlioMIP1 ensemble by Brierley (2015). Results for individual models are included in the Supplement.

We assess the properties of ENSO variability using the second, third and fourth statistical moments of the Niño3.4 index,

namely the standard deviation (s.d.), skewness and kurtosis. The first moment, the mean, is zero by definition. The ENSO amplitude is defined as the s.d. of the Niño3.4 time series. Furthermore, the skewness and kurtosis of the Niño3.4 index are computed (following the Fischer-Pearson definition (Zwillinger and Kokoska, 2000)). Both are normalised on the variance and a factor 3.0 is subtracted from the kurtosis to give 0.0 kurtosis for a normal distribution. Positive or negative skewness provides information on whether there are more El Niño or more La Niña events, respectively. Positive kurtosis indicates that

there are more values around the mean and the s.d. is mainly determined by extreme values, while negative kurtosis implies a more uniform distribution of values. The kurtosis thus provides information on the relative occurrence of extreme events (i.e. El Niño and La Niña events). The variation of the Niño3.4 s.d. has been investigated using 500 years of data from CESM2 and





**Table 2.** Correlation coefficient of the first principical component (PC1) of SSTs in the tropical Pacific to the different Niño indices. The largest correlation coefficient for each model is shown in **bold text**.

| Model | $E^{280}$: correlation of PC1 with | | | | $Eoi^{400}$: correlation of PC1 with | | | |
|---|---|---|---|---|---|---|---|---|
| | Niño4 | Niño3.4 | Niño3 | Niño1+2 | Niño4 | Niño3.4 | Niño3 | Niño1+2 |
| CCSM4 | 0.94 | **0.99** | 0.98 | 0.90 | 0.95 | **0.96** | 0.95 | 0.80 |
| CCSM4-UoT | **0.97** | **0.97** | 0.95 | 0.81 | **0.95** | **0.95** | 0.92 | 0.77 |
| CCSM4-Utr | 0.95 | **0.99** | 0.97 | 0.89 | 0.88 | **0.89** | 0.88 | 0.67 |
| CESM1.2 | 0.96 | **0.98** | 0.96 | 0.78 | 0.96 | **0.97** | 0.93 | 0.77 |
| CESM2 | 0.95 | **0.97** | 0.96 | 0.90 | 0.95 | **0.98** | 0.96 | 0.83 |
| COSMOS | **0.95** | **0.95** | 0.92 | 0.83 | **0.97** | **0.97** | 0.93 | 0.86 |
| EC-Earth3.3 | 0.90 | **0.96** | 0.95 | 0.76 | 0.87 | **0.90** | **0.90** | 0.68 |
| GISS2.1G | 0.89 | **0.99** | **0.99** | 0.93 | 0.82 | 0.98 | **0.99** | 0.91 |
| HadCM3 | **0.95** | 0.93 | 0.88 | 0.75 | **0.86** | 0.76 | 0.70 | 0.58 |
| HadGEM3 | 0.92 | **0.97** | 0.96 | 0.72 | 0.93 | **0.96** | 0.92 | 0.62 |
| IPSLCM5A | 0.93 | **0.95** | 0.93 | 0.74 | 0.93 | **0.94** | 0.92 | 0.76 |
| IPSLCM5A2 | **0.96** | **0.96** | 0.93 | 0.76 | 0.94 | **0.95** | 0.92 | 0.75 |
| IPSLCM6A | 0.94 | **0.98** | 0.95 | 0.78 | 0.96 | **0.97** | 0.95 | 0.74 |
| MIROC4m | **0.91** | 0.89 | 0.82 | 0.70 | 0.82 | **0.89** | 0.87 | 0.71 |
| MRI2.3 | 0.91 | **0.95** | 0.91 | 0.77 | 0.94 | **0.96** | 0.95 | 0.84 |
| NorESM-L | 0.92 | **0.97** | 0.93 | 0.75 | 0.85 | **0.92** | 0.91 | 0.69 |
| NorESM1-F | 0.87 | 0.94 | **0.97** | 0.75 | 0.89 | **0.95** | 0.91 | 0.70 |
| Ensemble mean | 0.93 | **0.96** | 0.94 | 0.80 | 0.91 | **0.94** | 0.91 | 0.75 |
| HadISST 1920 - 2020 | 0.88 | **0.96** | **0.96** | 0.77 | - | - | - | - |

MIROC4m and was found to not be large enough to impact the conclusions of the ensemble. Results of this are included in the Supplement.

To investigate the ENSO period(s), we perform a spectral analysis of the Niño3.4 index. Before the power spectra are computed, the time series are standardised using their s.d. It was chosen to use the multi-taper spectral method as explained by Ghil et al. (2002), with 3 tapers and a bandwidth parameter of 2. This spectral method is preferred over the classic periodogram because of its statistical robustness and reduction of spectral leakage. All spectra are scaled with respect to their respective sum. We assess the significance of spectral peaks by performing a red noise test on each time series. 90%, 95% and 99% confidence

levels are determined using a first order autoregressive (AR(1)) model with 10,000 Monte-Carlo generated surrogates.

     To quantify a change in ENSO period in the mid-Pliocene from the pre-industrial, we consider two measures: 1. The ensemble mean power spectra, and 2. The periods of the ensemble sum of peaks that are above the 90%, 95% and 99% confidence levels. The mean power spectrum gives information on what happens to the spectral power in the mid-Pliocene. Note that the mean is taken per frequency bin to create the mean spectrum for the $Eoi^{400}$ and $E^{280}$ runs. However, a mean spectrum does not



contain any information on the significance of the spectral peaks. For this reason, we introduce a procedure where we count all the spectral peaks above the 90%, 95% and 99% confidence levels in the 1.5 - 10 year period per simulation. The periods of these significant peaks are then binned for the whole ensemble. This histogram of the significant peaks provides information on the period or period range that represents the most ENSO-related activity. The method has been tested for robustness by repeating the procedure using 5 instead of 3 tapers; using a bandwidth parameter of 4 instead of 2; using the classic periodogram

(fast fourier transform) instead of the multi-taper method; and using the Niño3 instead of the Niño3.4 index. The conclusions are found to be independent of these different settings. A summary of these results is included in the Supplement.

We use principal component analysis (PCA) on the monthly SST anomalies to investigate changes in the spatial structure of ENSO. The first empirical orthogonal function (EOF) is determined in the tropical Pacific (defined as 23°S - 23°N and 140°E - 80°W). We follow the methodology of Power et al. (2013) in performing EOF analysis and normalisation, as did Brierley

(2015) for the PlioMIP1 ensemble. As the climatology and trend are removed, we expect the first EOF in the tropical Pacific to correlate with ENSO variability and not with the seasonal cycle. The spatial EOF pattern and the percentage of variance explanation will be compared for both simulations, to assess to which degree ENSO variability differs.

We can distinguish between two El Niño 'flavours', being 1. the Cold Tongue or East Pacific El Niño and 2. the Warm Pool or Central Pacific El Niño (also known as El Niño Modoki, Ashok and Yamagata (2009); Yeh et al. (2009)). These types

are distinguishable based on the region of their largest ENSO amplitude (hence the naming). Here we use the methodology proposed by Ren and Jin (2011), using two new indices for the Cold Tongue ($N_{CT}$) and Warm Pool ($N_{WP}$) El Niño, combining the Niño3 ($N_3$) and Niño4 ($N_4$) indices:

$$N_{CT} = N_3 - \alpha N_4,$$
$$N_{WP} = N_4 - \alpha N_3,$$

where $\alpha = 2/5$ if $N_3 N_4 > 0$ and $\alpha = 0$ otherwise. We will quantify if there is a change in El Niño flavour in the mid-Pliocene by computing and comparing the s.d. of the $N_{CT}$ and $N_{WP}$ indices.

Lastly, we will look at the tropical Pacific mean state, specifically the zonal SST gradient in the equatorial Pacific, to find explanations for a change in ENSO behaviour. For this, we compute the annual mean SSTs in the tropical Pacific (23°S - 23°N and 140°E - 80°W) from the 100 years monthly data for each model and simulation. We define the zonal SST gradient as the

SST difference between the warm pool (5°S - 5°N, 150°E - 170°E) and cold tongue (5°S - 5°N, 120°W - 100°W) regions. Furthermore, to assess whether the mean state of the mid-Pliocene becomes more 'El Niño-like', we spatially correlate the annual mean SST change (Eoi$^{400}$ - E$^{280}$) to the pre-industrial (E$^{280}$) pattern of the leading EOF.

### 2.2.3  Comparison with observations

We compare pre-industrial simulation results to those obtained via HadISST observational data, taken from 1920-2020 (Rayner

et al. (2003), latest data available through https://www.metoffice.gov.uk/hadobs/hadisst/data/download.html, last access: 11 May 2021). The observational data can be used to assess how well individual models as well as the pre-industrial ensemble





mean perform with respect to historical ENSO records. Note that there will be impacts of anthropogenic forcing such as greenhouse gas emissions in the HadISST observational data, that will not be present in the pre-industrial simulation results.

The mid-Pliocene mean climate simulation results will be compared with a reconstructed SST proxies by Foley and Dowsett

(2019) and McClymont et al. (2020). These reconstructions resemble the 30,000 years around the KM5c interglacial in the mid-Pliocene that the Eoi$^{400}$ simulations are designed to represent. The Foley and Dowsett (2019) data, part of PRISM4, is a collection of alkenone paleothermometry and U$_{37}^{k'}$ SST proxies. The McClymont et al. (2020) data is a combination of U$_{37}^{k'}$ reconstructions using a BAYSPLINE calibration and foraminifera Mg/Ca SST reconstructions (https://doi.pangaea.de/10.1594/PANGAEA.911847). Together these proxies represent eight SST values on six different locations in the tropical Pacific.

## 3 Results


### 3.1 ENSO variability

#### 3.1.1 Moments

Figure 2 shows the s.d., skewness and kurtosis for all models with respect to their pre-industrial values. The ensemble mean is also shown in the plots and the HadISST observational data is included as reference. Individual Niño3.4 index time-series for

all ensemble members are included in Supplement Figure S1.

Figure 2(a) shows a clear reduction of the Niño3.4 s.d. in the Eoi$^{400}$ ensemble mean compared to the E$^{280}$ ensemble mean, with 15 out of 17 individual models agreeing. The mid-Pliocene hence shows a 24% reduction in ENSO amplitude (compared to 20% in PlioMIP1, Brierley (2015)). The spread on the values is large; the difference between the smallest and largest s.d. is around 1.2 °C for both the pre-industrial and mid-Pliocene simulations. Still, the ensemble mean pre-industrial value is

close to the HadISST observations (0.91±0.28 °C and 0.77 °C, respectively). Notable individual models are COSMOS for having large s.d. in both runs, and CCSM4-Utr for showing the largest absolute decrease in s.d. The models showing almost no difference between E$^{280}$ and Eoi$^{400}$ are COSMOS, GISS2.1G and MRI2.3. The same model version of MRI was included in the PlioMIP1 ensemble, where it also showed no change (Brierley, 2015).

A less coherent pattern arises from the skewness of the Niño3.4 index, shown in Figure 2(b). Again, the spread of the values

is large. Most individual models (14 out of 17), as well as the ensemble mean, show a smaller skewness in the pre-industrial compared to the HadISST observations. The ensemble mean shows no significant change of the skewness in the mid-Pliocene simulations. Around half of the models (9 out of 17) show a positive skewness in both simulations, indicating generally more El Niño than La Niña events. However, the individual models do not agree to what happens to the skewness in the Eoi$^{400}$ runs, compared to the E$^{280}$ reference. GISS2.1G shows the most negative skewness in the mid-Pliocene (also in PlioMIP1).

Both NorESM-L and EC-Earth3.3 show the most negative skewness in the pre-industrial. Notably again is CCSM4-Utr for having the largest skewness in the pre-industrial and the biggest absolute decrease in the mid-Pliocene. CESM1.2 shows the opposite behavior with the largest increase of the skewness in the mid-Pliocene, although the pre-industrial result shows a good agreement with the HadISST data.



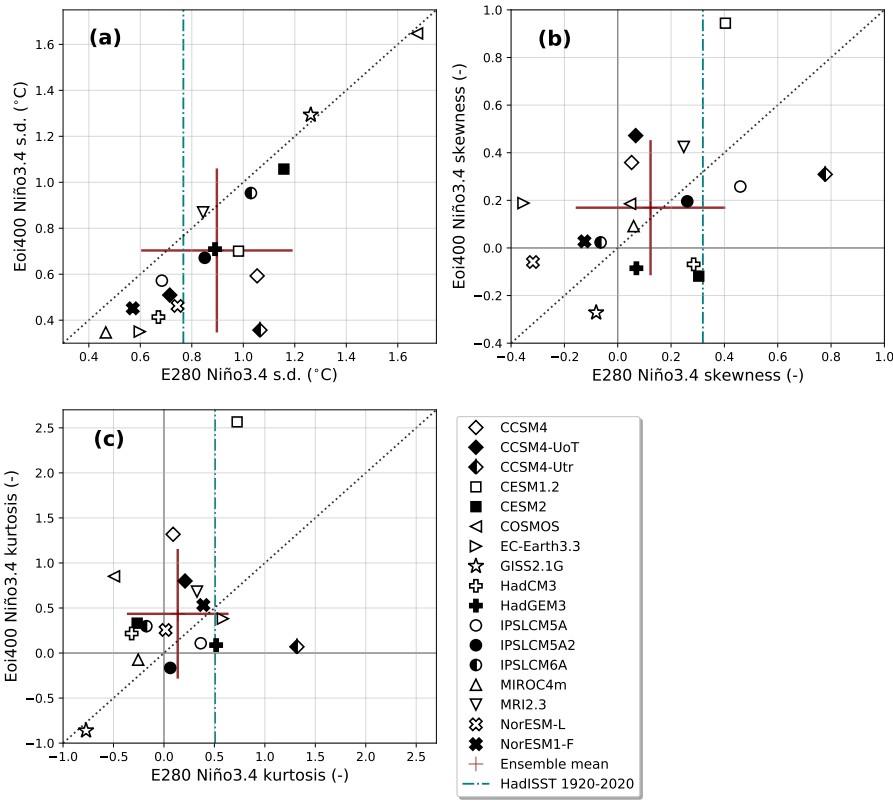

**Figure 2.** Niño3.4 moments for the $E^{280}$ and $Eoi^{400}$ runs; **(a)** standard deviation (s.d.), **(b)** skewness and **(c)** kurtosis for all models. Ensemble mean indicated as cross, the width and height indicate ensemble standard deviation. Dashed vertical line indicates moments from the HadISST Niño3.4 index.

The kurtosis of the Niño3.4 index is shown in Figure 2(c). The pre-industrial ensemble mean kurtosis is close to zero, which
does not agree with results from the HadISST observations (albeit within standard deviation range), that shows a positive kurtosis. The ensemble mean shows an increase in kurtosis in the mid-Pliocene simulations, with 11 out of 17 individual models agreeing. This indicates that the distribution becomes more 'heavy-tailed', implying that there is more activity around the mean (neutral state) and the variance is mainly determined by some extreme tail-values (i.e. 'extreme' El Niño or La Niña events). The notable deviations from the ensemble are GISS2.1G (large negative kurtosis in both runs), CCSM4-Utr (largest
decrease) and CESM1.2 (largest increase).

### 3.1.2 Spectral analysis

We investigate the ENSO period by looking at the Niño3.4 power spectra. The power spectra for all the individual simulations are shown in Supplement Figure S2, including the HadISST spectrum. In order to quantitatively check what happens in the ensemble, we look at the ensemble mean power spectrum and the histogram of significant peaks.





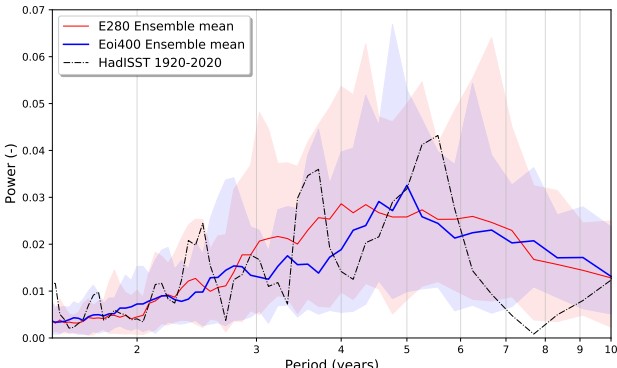

**Figure 3.** Ensemble mean Niño3.4 power spectra for the pre-industrial E$^{280}$ and mid-Pliocene Eoi$^{400}$ runs. For reference, the HadISST observational Niño3.4 spectrum is included. Shading represents a range of modelled values, specifically the second to smallest and second to largest values.

The ensemble mean power spectra are shown in Figure 3. The HadISST spectrum is also included. The shading represents the second to smallest and second to largest power value per frequency. The pre-industrial spectrum resembles the HadISST spectrum, except for a less clear separation of the spectral peaks. The pre-industrial mean spectrum shows the largest power in the 3 to 7 year period, that is generally associated with ENSO. The mid-Pliocene mean spectrum shows a peak at a period of 5 years. The clearest change in the mid-Pliocene is the reduction of power in the 3 - 4 year period. However, the range in modelled
values is large for both simulations, and the individual power spectra show a lot of differences in power and peak locations (see Supplement Figure S2). Some of the models show little differences between the pre-industrial and mid-Pliocene simulations (for example CCSM4). Many show a decrease of spectral intensity in the mid-Pliocene. However, GISS2.1G and MRI2.3 also show significant peaks (>99% CI) in the mid-Pliocene simulation that are not present in the pre-industrial reference. The clear shift to longer periods or lower frequencies found in the PlioMIP1 ensemble mean spectrum, is not reproduced by the PlioMIP2
ensemble mean (Brierley, 2015).

A way to assess the change in ENSO period in the mid-Pliocene is through counting the significant peaks in the spectra, as explained in Section 2.2. Figure 4 shows (a) a histogram of the number of peaks that are above a certain (90%, 95% and 99%) confidence level per period bin and (b) the difference between the mid-Pliocene and pre-industrial. It can be seen from Figure 4(a) that the majority of the significant peaks in the pre-industrial are found in the 2 - 5 year period, with a clear maximum
around the 3 - 4 year period. The total number of spectral peaks that can be called significant decreases in the mid-Pliocene for all the confidence levels. The number of peaks that are above the 99% confidence level reduces by one third.

The change in number of significant peaks is shown in Figure 4(b). It clearly shows the decrease in the number of peaks for all the confidence levels, mainly in the 2 - 5 year period. The reduction is strongest in the 3 - 4 year period. In the 1.5 - 2 year period, there is a slight increase in the number of peaks above the 90% and 95% confidence levels. The change in number of





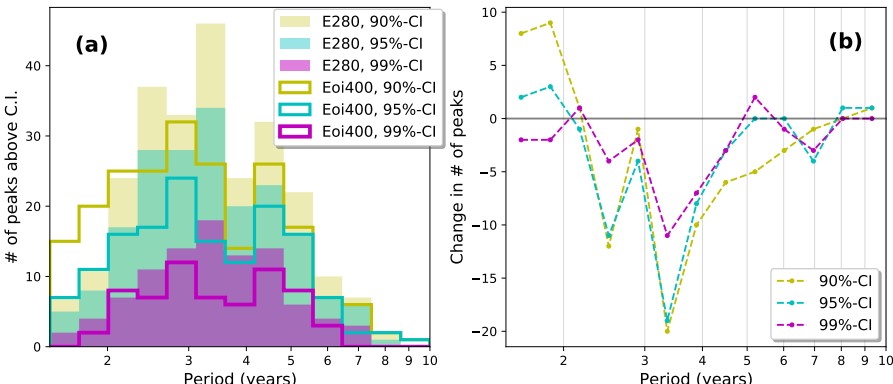

**Figure 4.** Histogram of the number of spectral peaks that are above the 90%, 95% and 99% confidence levels, binned in the 1.5-10 year period range. The number of peaks are taken from the multitaper spectra of the Niño3.4 index for **(a)** pre-industrial E$^{280}$ and mid-Pliocene Eoi$^{400}$ runs and **(b)** difference between the two.

peaks mainly shows that the spectral power in the 3 - 4 year El Niño period significantly decreases and that this power does not shift to different periods.

### 3.1.3 Spatial structure

To study the changes in spatial structure of ENSO, we compute the EOFs of the tropical Pacific SST anomalies. The first EOFs for all the individual simulations are shown in the Supplement Figure S3. The EOFs are normalised to be positive in the Niño3.4 region. The percentage of the variance that is explained by the first EOF is also computed and shown in the bottom left of the plots.

To compare the mid-Pliocene to the pre-industrial EOFs, the ensemble means are shown in Figure 5(b-d), also including the 1st EOF of the HadISST anomalies in 1920-2020 in (a). The stippling is included in regions where less than 12 out of 17 ($\sim 70\%$) models agree on the sign of the EOF. It can be seen from Figure 5(b) and (c) that the spatial structure looks qualitatively the same in both simulations. Also, the majority of the individual models agree with the ensemble mean pattern, indicated by the spread of the stippling. The majority of the individual models show little change, with only CCSM4-Utr, HadCM3 and MIROC4m showing spatial differences. The percentage of variance in the tropical Pacific that the first EOF explains, decreases from 50% in the pre-industrial to 42% in the mid-Pliocene within the ensemble mean. 14 out of 17 models agree with this decrease, with only MRI2.3 showing a slight increase in the mid-Pliocene run and COSMOS and GISS2.1G showing no change. The percentage of variance explained is very similar in the pre-industrial ensemble mean and the HadISST result.

The difference between the two ensemble mean EOFs is shown in Figure 5(d) and shows a decrease in the central equatorial Pacific region. This is in agreement with the ensemble mean reduction in ENSO amplitude (or Niño3.4 s.d.). The EOF difference shows an increase in the region slightly to the north and south of this region. This suggests that there is a shift of the El





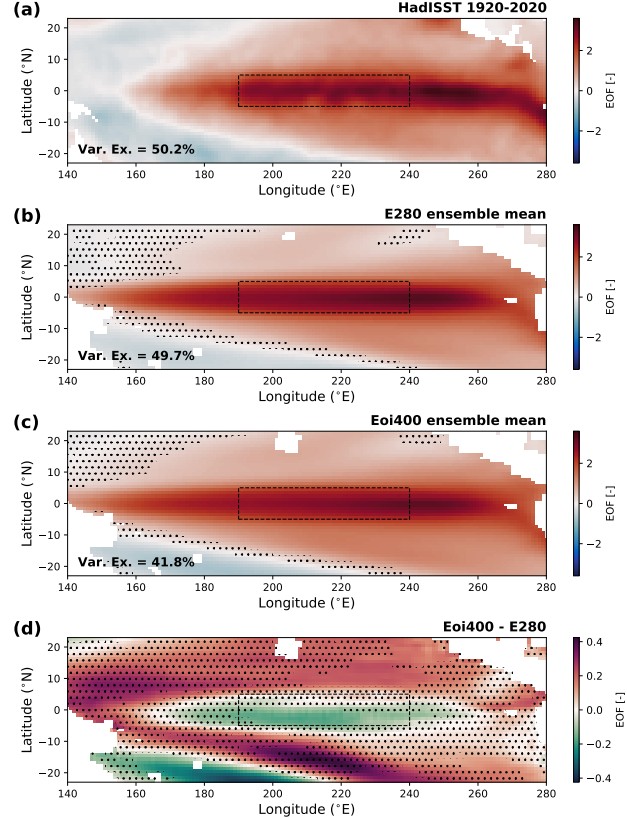

**Figure 5.** The ensemble mean empirical orthogonal function (EOF) computed for the tropical Pacific (region shown). Ensemble mean percentage that explains the variability in the region is shown in the bottom left. **(a)** Results for HadISST 1920-2020 observations, **(b)** E$^{280}$, **(c)** Eoi$^{400}$ and **(d)** difference. Stippling is included if less than 12 out of 17 ($\sim 70\%$) models agree with the sign of the EOF (difference). Dashed region indicates the Niño3.4 region, for reference.

Niño warmth expanding further across the tropical Pacific. This is also concluded in the PlioMIP1 ensemble, where the spatial pattern of the EOF difference looks qualitatively very similar (Brierley, 2015). Accompanying this meridional expansion of the warmth of an El Niño event is the southward migration of the South Pacific Convergence Zone (SPCZ), as was also shown by Pontes et al. (2020). It can be seen from the stippling that the majority of the models agree with a decrease of the EOF signal in the Niño3.4 region. The rest of the tropical Pacific shows large areas with stippling, indicating that there is a lot of model disagreement on the sign of the change in EOF.

The correlation coefficient of the leading principal component with the different Niño indices is shown in Table 2. As was concluded before, the majority of the models have the largest correlation in the Niño3.4 region, both in the E$^{280}$ as well as in the Eoi$^{400}$ simulations. This result agrees with the EOF pattern results showing that the area in the tropical Pacific with the largest ENSO variability is not significantly different in the mid-Pliocene.





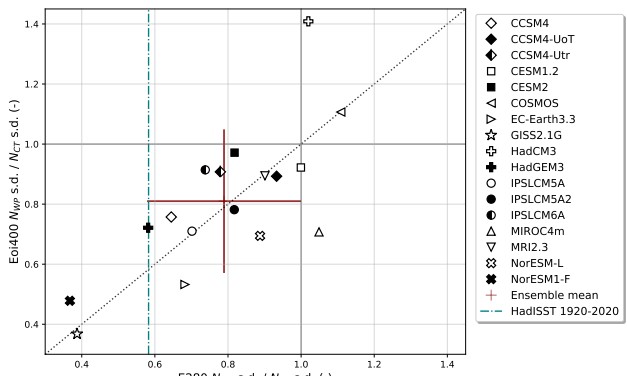

**Figure 6.** Ratio of WP El Niño index ($N_{WP}$) s.d. to the CT El Niño index ($N_{CT}$) s.d. for the pre-industrial $E^{280}$ and mid-Pliocene $Eoi^{400}$ simulations. Ensemble mean indicated as cross, the width and height indicate ensemble standard deviation.

### 3.1.4 El Niño flavour

A change in El Niño 'flavour' in the pre-industrial and mid-Pliocene simulations will be investigated using the methodology by Ren and Jin (2011). The amplitude of the cold tongue or east Pacific El Niño is defined as the standard deviation (s.d.) of the $N_{CT}$ index. Likewise, for the warm pool or central Pacific El Niño the s.d. of the $N_{WP}$ index is used. The change in magnitude of both amplitudes follows the changes in the Niño3.4 s.d. results. We are interested in the difference in the ratio between the two types. The ratio of the warm pool El Niño amplitude to the cold tongue El Niño amplitude $N_{WP}/N_{CT}$ is shown in Figure 6. For the majority of the models (13 out of 17), the ratio is smaller than 1 in both of the simulations, implying there are generally more cold tongue El Niño events. This is in agreement with the result from the HadISST observations, although the HadISST ratio is smaller than most of the model results. The ensemble mean shows no change in the ratio of warm pool to cold tongue El Niño events, and most individual models also show little change. Notable is HadCM3 for showing a majority and large increase in the number of warm pool El Niño events in the mid-Pliocene, and NorESM1-F and GISS2.1G for showing a small amount of warm pool El Niño events in both simulations. Yeh et al. (2009) present a clear increase in the ratio (>1) for a future scenario (fixed 700 ppmv $CO_2$), which does not agree with the results of the PlioMIP2.

### 3.2 Tropical Pacific mean climate

In this section, we will analyse changes in the tropical Pacific mean state, and investigate if this relates to the changes in ENSO variability in the mid-Pliocene simulations. We look at the tropical Pacific annual mean SSTs and specifically the zonal SST gradient along the equatorial Pacific, and relate these to changes in the Niño3.4 s.d. and the pre-industrial leading EOF.



### 3.2.1 Annual mean pattern

Figure 7 shows the annual mean SST patterns in the tropical Pacific (23°S - 23°N, 140°E - 280°E). The figure shows (a) results for the HadISST data from 1920-2020 (no detrending), (b-c) the pre-industrial and mid-Pliocene ensemble means and (d) the ensemble mean differences. The circles in (b) represent the HadISST values on selected proxy locations. It can be seen that the $E^{280}$ ensemble mean pattern looks qualitatively similar to the HadISST observations. Consistent with other studies using coupled GCMs (Collins et al., 2010; Coats and Karnauskas, 2017; Brown et al., 2020), the $E^{280}$ means shows colder temperatures in the central and west Pacific and slightly warmer temperatures along the east Pacific coast compared to observations. It should be noted that this 'cold bias' can also be expected since we are comparing equilibrium pre-industrial simulations with historical observations.

The $Eoi^{400}$ ensemble mean is warmer but qualitatively similar to the pre-industrial result. The markers in Figure 7(c) are SST proxies reconstructed by Foley and Dowsett (2019) (hereafter referred to as PRISM4) and by McClymont et al. (2020), including two $U^{k'}_{37}$ and two Mg/Ca reconstructions (hereafter referred to as MC-UK37 and MC-Mg/Ca, resp.). There is good agreement between the PlioMIP2 ensemble mean and the PRISM4 SSTs in the East Pacific upwelling region. The two points in the warm pool region show slightly lower SSTs compared to the $Eoi^{400}$ ensemble mean. The MC-UK37 point around 240°E shows a higher value compared to the ensemble mean, while the MC-Mg/Ca around 273°E shows a lower value.

Figure 7(d) shows the ensemble mean $Eoi^{400}$ - $E^{280}$ difference. The colorbar of the plot highlights deviations from the tropical Pacific mean SST difference (∼ +1.9 °C), meaning that the red and green areas show warmer and cooler parts with respect to the tropical Pacific mean difference, respectively. It shows specific warmer parts along the equator and in the upwelling region in the East Pacific. It also shows a clear cooler part in the southern tropical Pacific, implying a shift of the SPCZ as is also seen from the change in EOF (Figure 5(d)). This result agrees with findings by Pontes et al. (2020) that investigated shifts in precipitation patterns in the PlioMIP1 and PlioMIP2 ensembles.

The six points (**P-U**) represent differences between proxies and HadISST, the respective eight values are presented in Table 3. Points **P** and **Q** in the warm pool region show similar temperatures as in the pre-industrial, not agreeing with the increase in temperature seen in from the PlioMIP2 ensemble mean. Points **R-U** in the central and east Pacific show a better agreement with the ensemble mean difference, albeit with a range of ±1 °C. The only clear outlier is the MC-Mg/Ca proxy at 273°E (at point **T**), indicating a significant SST decrease compared to the pre-industrial. This temperature decrease is not captured by the PlioMIP2 ensemble mean difference nor by any individual model.

### 3.2.2 Zonal SST gradient

Early observational work suggested a highly reduced zonal SST gradient in the tropical Pacific in the mid-Pliocene (Wara et al., 2005; Fedorov et al., 2013) and it has been argued that ENSO properties might be related to this mean state feature. The PlioMIP1 ensemble did not show a clear agreement on the zonal SST gradient, and it was concluded that a reduction in zonal SST gradient could not explain a reduction in ENSO variability (Brierley, 2015). Here, we will investigate if there is any correlation between the zonal SST gradient and the change in Niño3.4 s.d. in PlioMIP2.




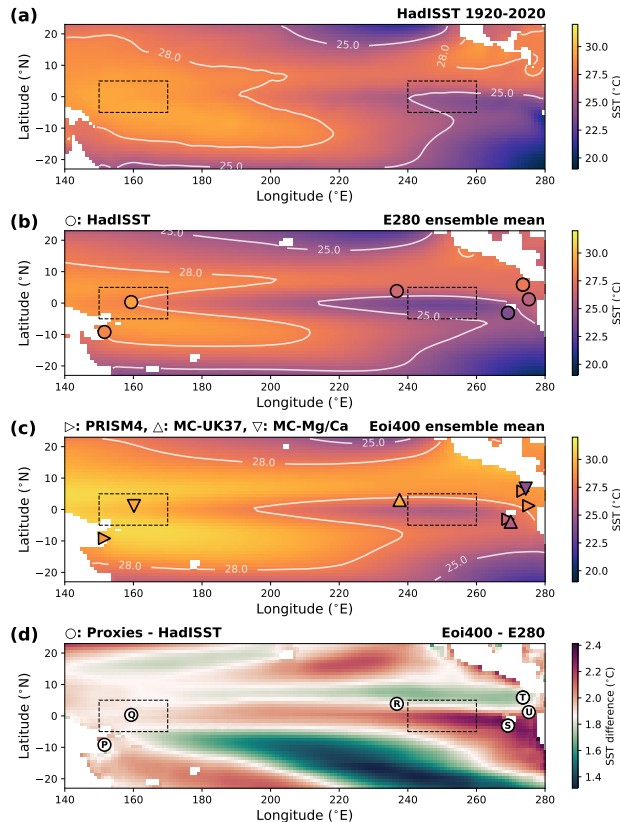

**Figure 7.** The ensemble mean annual mean SSTs in the tropical Pacific. **(a)** Results for HadISST 1920-2020 observations, **(b)** E$^{280}$, **(c)** Eoi$^{400}$ and **(d)** difference between Eoi$^{400}$ - E$^{280}$. Dashed regions are the west Pacific warm pool and east Pacific cold tongue, the zonal SST gradient is computed as a difference of the mean SST in these regions. Points in **(b)** show the HadISST values on the proxy locations and **(c)** the reconstructed mid-Pliocene SST proxies from PRISM4 and MC. Points in **(d)** show the difference between the proxies and the HadISST data, values are presented in Table 3.

Figure 8 shows the ensemble mean, meridional mean (5°S - 5°N) SST in the Pacific Ocean. The individual model results are included in Supplement Figure S4. The HadISST 1920-2020 result is included for reference. The shading shows the range of modelled values. The HadISST values fall within the range of pre-industrial modelled values but are about 1 °C warmer. This can be expected since the HadISST data includes recent global warming trends. The mid-Pliocene ensemble mean follows the same zonal dependence as the pre-industrial but is consistently 2 °C warmer, as can be seen from the Eoi$^{400}$ - E$^{280}$ difference.

However, the model spread is large. The range of values is especially large in the East Pacific cold tongue region, where the modelled values range from 0 to 4 °C warming.

We quantify the zonal SST gradient in the tropical Pacific as the SST difference between the West Pacific warm pool (5°S - 5°N, 150°E - 170°E) and East Pacific cold tongue (5°S - 5°N, 120°W - 100°W) regions. These regions are also indicated in Figures 7 and 8. Figure 9(a) shows the zonal SST gradients for the pre-industrial and mid-Pliocene simulations. The ensemble





**Table 3.** SST difference on six different locations in the tropical Pacific, both mid-Pliocene proxy minus HadISST 1920-2020 results as well as $Eoi^{400}$ minus $E^{280}$ difference. Proxy reconstructions by Foley and Dowsett (2019) (part of PRISM4) and McClymont et al. (2020) (referred to as 'MC'). Locations also indicated in Figure 7(d).

| # | Longitude (°E) | Latitude (°N) | Source | Proxies - HadISST (°C) | $Eoi^{400}$ - $E^{280}$ (°C) |
|---|---|---|---|---|---|
| P | 152 | -9.2 | PRISM4 | 0.50 | 1.78 |
| Q | 159 | 0.3 | MC-Mg/Ca | 0.13 | 1.89 |
| R | 236 | 3.8 | MC-UK37 | 3.1 | 1.91 |
| S | 269 | -3.1 | PRISM4 | 2.6 | 2.24 |
| | | | MC-UK37 | 1.7 | 2.24 |
| T | 273 | 5.8 | PRISM4 | 0.45 | 1.75 |
| | | | MC-Mg/Ca | -3.6 | 1.75 |
| U | 275 | 1.2 | PRISM4 | 3.0 | 2.01 |

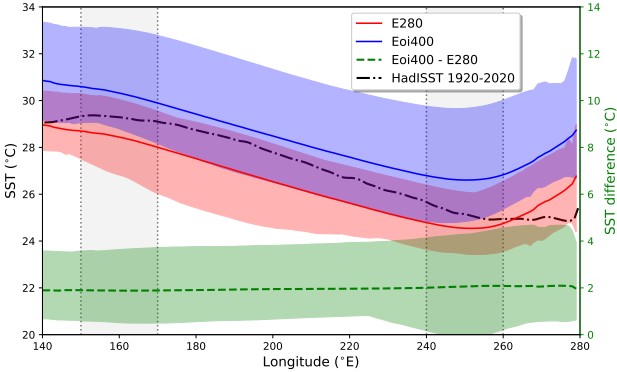

**Figure 8.** Ensemble mean equatorial Pacific SSTs (5°S - 5°N mean), difference shown as dashed green line (following right vertical axis). HadISST 1920 - 2020 observation included as dashed-dotted line. Coloured shaded area shows the range of modelled values. Grey vertical rectangles show the region considered here to be the warm pool (left) and cold tongue (right).

mean shows little change in the mid-Pliocene, and a reasonable agreement with the HadISST observations. This result is in line with what is shown in Figure 8. However, a majority of the models (13 out of 17) actually shows a slightly reduced zonal SST gradient in the mid-Pliocene simulations. The ensemble mean is very much affected by two outliers, namely MRI2.3 and COSMOS, that show a greatly increased zonal SST gradient. It should be noted that the pre-industrial simulation of MRI2.3 has its cold tongue centred around 140°W instead of 110°W, and this is the main reason for the large difference in the zonal

SST gradient (see the individual model result in Supplement Figure S4).

To investigate the relation between the zonal SST gradient and ENSO amplitude, we present a scatter plot of the change in zonal SST gradient versus the change in Niño3.4 s.d. in Figure 9(b). The ensemble mean shows a $\sim 24\%$ decrease in Niño3.4 s.d. despite little change in zonal SST gradient (-0.17 ± 0.77 °C). The 'cluster' of models around the ensemble mean does show





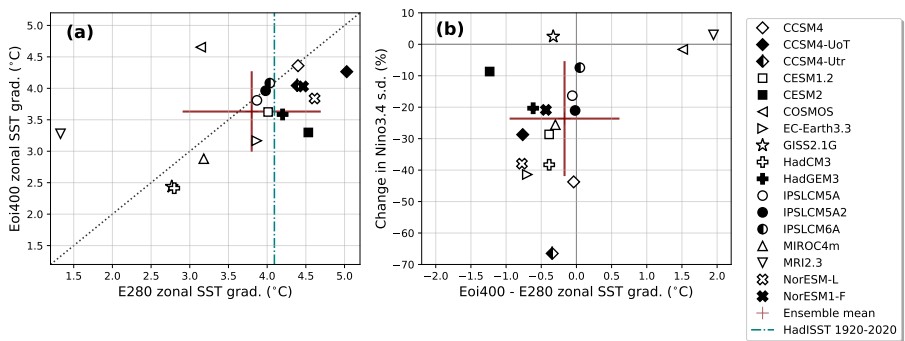

**Figure 9. (a)** Scatter plot of $E^{280}$ versus $Eoi^{400}$ zonal SST gradient, red cross represents ensemble mean, dashed-dotted line shows the HadISST 1920-2020 value. **(b)** Scatter plot showing the change in Niño3.4 s.d. as a function of the change in zonal SST gradient in the equatorial Pacific.

a slightly more reduced zonal SST gradient together with a robust reduction in Niño3.4 s.d. On the one hand, CESM2 shows

the largest reduction in zonal SST gradient but a small reduction in Niño3.4 s.d. On the other hand, COSMOS and MRI2.3 show a great increase in zonal SST gradient but a similar Niño3.4 s.d. - although we should consider the comment made on the MRI2.3 $E^{280}$ zonal SST gradient in the previous paragraph. CCSM4-Utr, however, shows the largest reduction in Niño3.4 s.d. despite a similar zonal SST gradient. Ultimately, there does not seem to be a strong correlation between the change in zonal SST gradient and the change in Niño3.4 s.d. in the PlioMIP2 ensemble. This conclusion is consistent with a recent study by

Brown et al. (2020) considering a large ensemble of CMIP5/6 and PMIP3/4 simulations.

### 3.2.3 The 'El Niño-like' mean state

To investigate if the mean SST pattern in the tropical Pacific is 'El Niño-like', we project the changes in the mean state (i.e. $Eoi^{400}$ SSTs - $E^{280}$ SSTs) onto the leading EOF of the pre-industrial $E^{280}$ simulation for each model and perform a spatial correlation. If this correlation is positive, it implies that the mid-Pliocene mean SSTs are more similar to the pre-industrial

El Niño than the pre-industrial mean SSTs. Likewise, if the correlation is negative, the mid-Pliocene mean can be said to be more similar to a La Niña. Results are shown in Figure 10(a). The ensemble mean shows no correlation of the mean state changes to either an El Niño or a La Niña, and about half of the individual models (8 out of 17) also show little correlation (between −0.3 and 0.3). Interestingly, there is a 'cluster' of models ("A": GISS2.1G, COSMOS, IPSLCM6A and CESM2) that show a clear positive correlation (El Niño-like) but little to no change in the Niño3.4 s.d. On the other side, there is cluster

- albeit less clearly grouped - showing a negative correlation (La Niña-like) and the largest reduction in Niño3.4 s.d. ("B": CCSM4-Utr, EC-Earth3.3, CCSM4). This contrasting result between CCSM4 and CESM2 was already observed by Feng et al. (2020). Taking into account all the models, there seems to be an increasing trend: the more 'El Niño-like', the smaller the reduction in ENSO amplitude is. A similar trend is found by Pontes et al. (2021, in prep.), when considering changes in the thermocline slope in the PlioMIP2 simulations. However, we should consider here the results from the PlioMIP1 ensemble





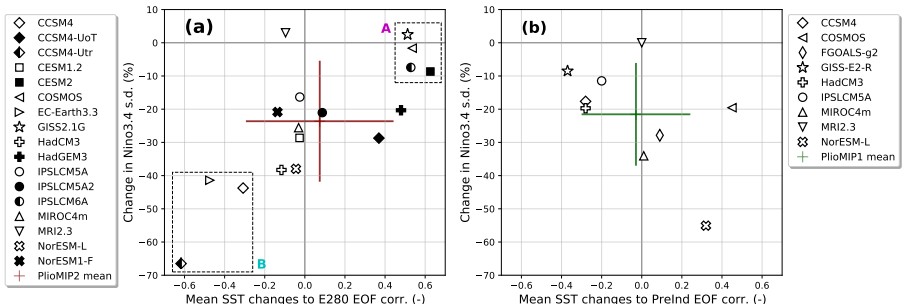

**Figure 10.** Scatter plot of the correlation coefficient of the annual mean SST changes to the leading pre-industrial EOF in the tropical Pacific versus the change in Niño3.4 s.d. **(a)** Result for the PlioMIP2 ensemble as well as **(b)** the PlioMIP1 ensemble (Brierley, 2015). Some models were a part of both ensembles, but note that a different protocol was used in the both MIPs.

(Brierley, 2015), shown in Figure 10(b), that suggest the opposite trend. Although some of the individual models that were also a part of PlioMIP1 show very different results in PlioMIP2, the ensemble mean is actually very similar, despite the fact that different boundary conditions are used in both MIP protocols.

To analyse the El Niño-like mean state in a more detailed fashion, we focus on the two 'clusters' that were identified in Figure 10(a). We show the cluster model-mean pattern of the annual mean SST changes and the pre-industrial EOF in the tropical Pacific in Figure 11. The first group of models is GISS2.1G, COSMOS, IPSLCM6A and CESM2 (group "A"), showing El Niño-like mean state changes and strong ENSO variability (Figure 11a-b). The mean state changes show a clear warming pattern along the equator, very much alike the pre-industrial EOF pattern. The warming is also very uniform, implying no changes in the zonal SST differences compared to the pre-industrial. The second group of models is CCSM4-Utr, EC-Earth3.3, CCSM4 (group "B") and show La Niña-like mean state changes with a strongly reduced ENSO amplitude (Figure 11c-d). The mean state changes show a relative weaker warming along the equator compared to the tropical Pacific mean SST changes. Large variations can be seen in the West Pacific, where the equatorial warm pool region warms the least and the subtropical regions show a higher than average warming. The mean state changes show a clear anticorrelation with the pre-industrial EOF pattern. Interestingly, this group also shows a large warming in the upwelling region along the South American Pacific coast. This implies a reduction in the zonal SST gradient between the upwelling region and the warm pool, in comparison to the pre-industrial.

## 4 Discussion

We have investigated ENSO variability in the PlioMIP2 ensemble using a set of different metrics. Some results are quite robust (reduced Niño3.4 s.d., similar spatial structure), but the ensemble often shows a large spread in values. In this section, we will discuss the results in light of observations (both present-day and palaeo), inter-model differences, and we will suggest possible physical mechanisms of a consistently reduced ENSO variability in the mid-Pliocene simulations.





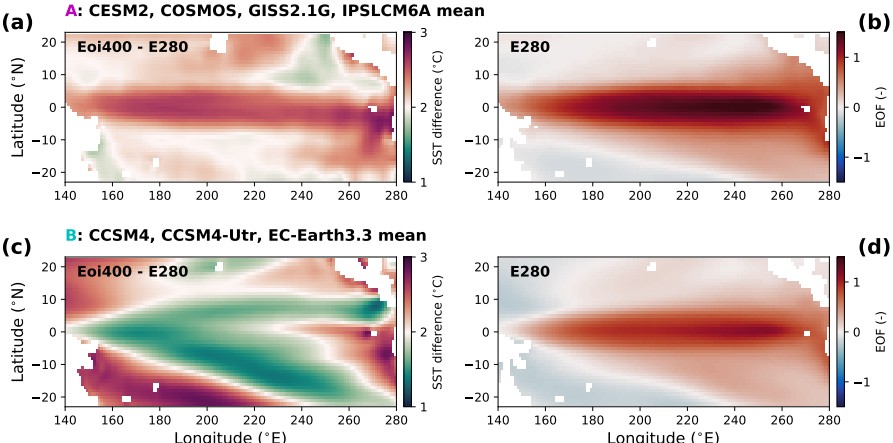

**Figure 11.** **(a, c)** 'Cluster'-mean annual mean SST changes and **(b, d)** pre-industrial leading EOF in the tropical Pacific. For **"A"**: CESM2, COSMOS, GISS2.1G and IPSLCM6A **(a, b)** and **"B"**: CCSM4, CCSM4-Utr and EC-Earth3.3 **(c, d)**.

## 4.1 Data-model comparison

### 4.1.1 Pre-industrial observations

We have included the results from the HadISST 1920 - 2020 dataset as a reference for the pre-industrial simulation results. The time range of the observational data does not cover the pre-industrial period and includes anthropogenic forcing trends (although a linear trend is removed). It was decided to use a 100 year time series, as with the PlioMIP2 data, and use the most recent data because of a higher spatial resolution. Chelton and Risien (2016) mention zonal and meridional discontinuities in the HadISST dataset, specifically in the Pacific. They advise caution for high-resolution studies, but deem the data adequate for large-scale SST variability studies. Haywood et al. (2020) prefer to use the NOAA ERSST v5 dataset (Huang et al., 2017) over the HadISST data because of consistency with other observational data-sets on a global scale.

The Niño3.4 moments of the $E^{280}$ simulations match reasonably well with those from the HadISST data (see Figure 2). The ensemble mean shows a slightly higher Niño3.4 s.d. (0.90) compared to the HadISST results (0.77). The PlioMIP1 ensemble mean reported a slightly lower s.d. (0.78, Brierley (2015)), and was compared to the Niño3.4 s.d. of the ERSST data (0.69). The overestimation of the model Niño3.4 s.d. compared to observational data is also reported in the CMIP5 ensemble (Kim et al., 2014). However, there is a good match with the HadISST tropical Pacific leading EOF and that of the $E^{280}$ ensemble mean; both in the percentage of variance explained (both 50%) and correlation of the Niño3.4 index with the first PC (both 0.96). This shows that the PlioMIP2 ensemble does a reasonable job in representing pre-industrial ENSO variability.

There is a slight mismatch considering the warm pool (or Central Pacific) to cold tongue (or East Pacific) El Niño occurrence ratio (Figure 6). The $E^{280}$ ensemble mean shows a considerably higher ratio (0.80) compared to the HadISST ratio (0.60). The



ensemble mean results do agree with a present-day model ensemble presented in Yeh et al. (2009) (approximately 0.6 - 0.9,
although a slightly different method was used).

### 4.1.2  Mid-Pliocene proxies

Figure 7 shows a reasonable agreement of the annual mean SSTs with the proxy reconstructions by Foley and Dowsett (2019)
and McClymont et al. (2020). The close agreement with most proxies in the East Pacific upwelling region is promising, as the
mid-Pliocene ensemble mean shows the greatest warming in this region compared to the pre-industrial ensemble mean. One
proxy around 240°E (point **R**) suggests a higher temperature than the ensemble mean. Considering the two proxy locations in
the western equatorial Pacific, the proxies suggest a flat zonal SST profile, possibly indicating an 'El Niño-like' mean state.
However, the ensemble mean shows more warming in the western equatorial Pacific and shows a clear zonal SST gradient (see
also Figure 8), albeit slightly reduced compared to the pre-industrial. It should be noted here that the Mg/Ca reconstructions
by McClymont et al. (2020) are reported to have a significant cold bias, in comparison to the $U_{37}^{k'}$ reconstructions and the
PlioMIP2 results. This can explain the severe underestimation in SST by the Mg/Ca proxy at 273°E, in comparison to the
PlioMIP2 ensemble mean. Another note regarding the McClymont et al. (2020) reconstructions is the reported error of 1.5 °C
on their data. This is approximately the same as the range of modelled values on the proxy locations. Taking into account these
uncertainties, there is a clear agreement between the observational differences (proxy minus HadISST) and model differences
as reported in Table 3, with the exception of the 273°E Mg/Ca proxy. Another comment is that some proxies are located close
to a coastal area, meaning that it could be affected by boundary currents that are not resolved with the ocean resolution of
most PlioMIP2 models. When putting the proxies in a global perspective, there is generally a good agreement with proxies
and the PlioMIP2 ensemble mean in the tropics, but discrepancies occur in the mid- and higher latitudes, specifically in the
Northern Hemisphere. This holds both for the PRISM4 proxies (Haywood et al., 2020) as well as for the McClymont et al.
(2020) proxies.

The PlioMIP2 ensemble shows that it is likely that there was a slight reduction of the zonal SST gradient in the mid-
Pliocene equatorial Pacific, but not as large as some earlier proxy reconstructions suggest. Furthermore, the mean state is not
specifically 'El Niño-like', as the individual models show a range of possibilities. The PlioMIP2 ensemble also shows clear
ENSO variability (although weaker than in the pre-industrial), agreeing with the proxy-based reconstructions by Scroxton
et al. (2011), Watanabe et al. (2011) and White and Ravelo (2020). More specifically, White and Ravelo (2020) suggest
that mid-Pliocene ENSO variability varied between reduced and similar to the present-day, which is also captured by the
PlioMIP2 ensemble. Watanabe et al. (2011), however, suggest that ENSO variability is similar to the present-day, something
most PlioMIP2 models do not agree with. Still, the PlioMIP2 ensemble includes four models (of which three participate in
CMIP6) that show both ENSO variability similar to present-day as well as a mean state that is more 'El Niño-like' (Figure
10(a)).





## 4.2 Model differences

The PlioMIP2 ensemble shows robust signals: the Eoi$^{400}$ simulations show a reduced ENSO variability (15 out of 17 models, Figure 2), a robust decrease of spectral peaks in the 3 - 4 year period (Figure 4), a similar spatial structure of ENSO compared to the pre-industrial (Figure 5 and Table 2) and a (slight) reduction of the tropical Pacific zonal SST gradient (14 out of 17 models, Figure 9). Still, the spread on the individual model results is often large, and not all findings are unambiguous. Below we will discuss the possible causes of differences between models.

### 4.2.1 Models deviating from the ensemble mean

Some of the clear outliers are:

- **(Niño3.4 index)** COSMOS for showing by far the largest Niño3.4 s.d. ($\sim$1.7°C) in both pre-industrial and mid-Pliocene simulations; COSMOS, GISS2.1G and MRI2.3 for showing almost unchanged Niño3.4 s.d. in both simulations; CCSM4-Utr for showing the largest reduction in Niño3.4 s.d. in the mid-Pliocene ($-67\%$); and GISS2.1G for showing a very regular Niño3.4 index and an almost single-peaked Eoi$^{400}$ power spectrum;

- **(ENSO structure)** CCSM4-Utr, HadCM3 and MIROC4m for showing large changes in the Eoi$^{400}$ EOF pattern; and COSMOS and HadCM3 for being the only models showing more Warm Pool to Cold Tongue El Niño events both in the pre-industrial and mid-Pliocene simulations;

- **(Mean SSTs)** COSMOS and MRI2.3 for showing a large increase in the zonal SST gradient from Eoi$^{400}$ to E$^{280}$ ($+1.5$ °C and $+2.0$ °C, resp.), where COSMOS shows a strong 'El Niño-like' mean state whereas MRI2.3 shows no particular correlation; CESM2 for showing the largest decrease in zonal SST gradient from Eoi$^{400}$ to E$^{280}$ ($-1.2$ °C), as well as a clear 'El Niño-like' mean state; and CCSM4-Utr for showing the most 'La Niña-like' mean state changes.

MRI2.3 did not use the provided 'enhanced' boundary conditions, but the 'standard' version (Haywood et al., 2016b). This is the best possible realization of the mid-Pliocene conditions (such as vegetation and land ice cover) using a modern land-sea mask with closed Arctic ocean gateways. This could explain why MRI2.3 shows an unchanged Niño3.4 index, as well as no change in the WP/CT El Niño ratio. Next to this, its Eoi$^{400}$ mean state changes show no particular correlation with the E$^{280}$ EOF, both in PlioMIP2 as well as in MRI2.3's contribution to PlioMIP1. However, also HadGEM3 used the pre-industrial land-sea mask for its mid-Pliocene simulations (with closed Canadian Archipelago, open Bering strait, mid-Pliocene vegetation and land ice cover), and clearly shows different ENSO variability in its both simulations. The explanation for MRI2.3's large increase in zonal SST gradient is its anomalous pre-industrial equatorial Pacific zonal SSTs, showing a cold tongue around 220 °E, disagreeing with the HadISST observations and all other ensemble members.

CCSM4-Utr started the Eoi$^{400}$ spin-up with ocean temperatures obtained from PlioMIP1 results, whereas most other models used pre-industrial conditions as ocean temperature initialization (Baatsen et al., 2021, in prep.). It shows one of the highest SAT differences, even though it has a very modest ECS compared to the rest of the ensemble (Haywood et al., 2020). Next to this, it shows a large increase in deep-ocean temperatures, namely 4 °C warmer (Baatsen et al., 2021, in prep.). This is high



considering the 2.8 °C SAT increase that the PlioMIP2 ensemble mean shows over oceans (Haywood et al., 2020). This could explain why CCSM4-Utr proves to be quite an outlier, especially showing a huge reduction in Niño3.4 s.d. and the most 'La Niña-like' mean state in the mid-Pliocene. However, also CESM1.2 and CESM2 have used a 'warm' ocean initialization (Feng et al., 2020) and do not show the same extreme behaviour.

COSMOS also stands out, and has one of the coarsest ocean resolutions (∼3°) of the ensemble. NorESM-L has a similar ocean resolution, however, but usually behaves quite close to the ensemble mean. COSMOS is furthermore the only model that included a dynamic vegetation model, but it seems unlikely that this could have such a big effect on tropical Pacific variability. COSMOS is the only non-CMIP6 model that has a high ECS (4.7 °C), although not the highest of the PlioMIP2 ensemble (Haywood et al., 2020).

Other models with a high ECS are HadGEM3 (5.5 °C), CESM2 (5.3 °C) and IPSLCM6A (4.8 °C) and are all CMIP6 models (Haywood et al., 2020; Williams et al., 2021). Where HadGEM3 and IPSLCM6A perform reasonably close to the ensemble mean, CESM2 values are often outside of the standard deviation range around the ensemble mean. It shows a Niño3.4 s.d. that is large both in the pre-industrial as well as the mid-Pliocene, and a change that is small compared to the ensemble mean (−9% to −24%). Furthermore, its mid-Pliocene mean state shows the largest reduction in zonal SSTs and the most 'El Niño-like' mean state. A high ECS alone cannot explain this. CESM1.2 generally performs close to the ensemble mean, and differences between CESM1.2 and CESM2 include an updated atmospheric model (CAM5 to CAM6), land model, sea ice model and a new mixing scheme in the ocean model (Feng et al., 2020), that could all explain the difference performance.

### 4.2.2 Model generation and family

PlioMIP2 includes CCSM4, CCSM4-UoT, CCSM4-Utr, CESM1.2 and CESM2 (the 'CESM' family) that share a lot of similarities, especially in their ocean models. The five models show a very similar pre-industrial Niño3.4 s.d. (∼0.8−1.2°C), but no agreement in the mid-Pliocene Niño3.4 s.d. (∼0.3−1.1°C). They do show a great similarity for both simulations in the ratio of WP and CT El Niño's. However, the five models show little agreement both in the change in zonal SST gradient as well as the 'El Niño-like' mean state results; CESM2 shows a strongly reduced zonal SST gradient and an 'El Niño-like' mean state, whereas CCSM4 shows almost no change in zonal SST gradient and a more 'La Niña-like' mean state. The results of CCSM4, CESM1.2 and CESM2 are discussed in more detail in Feng et al. (2020). They find a weakened Walker circulation and reduced upwelling in CESM1.2 and CESM2, but not in CCSM4, and this could explain the aforementioned differences.

The two NorESM models (NorESM-L and NorESM1-F) as well as the three IPSL model (IPSLCM5A, IPSLCM5A2 and IPSLCM6A) generally do show a great similarity between their results. HadCM3 and HadGEM3 both belong to the 'Hadley Centre' model family, but show very different results. Generally HadGEM3 performs close to both the pre-industrial and mid-Pliocene ensemble means. The HadGEM3 pre-industrial results furthermore show a great match with the HadISST observations. The HadCM3 results are overall less close to the ensemble mean. More information on the difference in large-scale features between HadCM3 and HadGEM3 can be found in Williams et al. (2021).

Brown et al. (2020) argue that the simulation of ENSO may be improved in CMIP6 (compared to CMIP5), especially due to improvements in the simulation of the mean state in the tropical Pacific, such as a reduced double intertropical convergence




zone (ITCZ) bias and cold tongue bias. When only the results of CMIP6 models (CESM2, EC-Earth3.3, GISS2.1G, HadGEM3, IPSLCM6A, NorESM1-F) are considered, the ensemble mean results and the conclusions on mid-Pliocene ENSO variability are similar. However, the CMIP6 models do show the strongest reduction in zonal SST gradient in the mid-Pliocene (Figure 9). Also, it is mainly the CMIP6 models that show 'El Niño-like' mean state changes (Figure 10), although EC-Earth3.3 exhibits a 'La Niña-like' mean state.

### 4.2.3  PlioMIP2 versus PlioMIP1

Seven of the seventeen models that participate in PlioMIP2 also participated in PlioMIP1, namely CCSM4, COSMOS, HadCM3, IPSLCM5A, MIROC4m, MRI2.3 and NorESM-L. Some of the PlioMIP1 conclusions presented in Brierley (2015) are similar: the Niño3.4 s.d. reduces (23% in PlioMIP2 versus 20% in PlioMIP1), the spatial structure of El Niño is similar and the El Niño flavour shows no robust change. However, there are also some clear differences. PlioMIP2 shows a significant reduction of the spectral power in the 3 - 4 year period, where PlioMIP1 mainly shows a shift of power to longer periods; the PlioMIP1 results showed a clear 'low frequency' peak in the pre-industrial around 5 - 6 years that seemed to shift to 7 - 8 years in the mid-Pliocene ensemble mean. The significance of the spectral peaks was not assessed, however. Another difference is that the PlioMIP2 models show a clear but small reduction in zonal SST gradient where PlioMIP1 shows 'no preferential warming in the Eastern equatorial Pacific'. An explanation for the difference between the ensemble results is that PlioMIP2 encompasses more and newer models (most CMIP6) that have a higher resolution and include the newest ocean/atmosphere models and parametrizations. Next to this, the boundary conditions of PlioMIP2 are more specifically tuned to a specific time slice in the mid-Pliocene (Dowsett et al., 2016). It should also be noted that Brierley (2015) used more than 100 years data (up to 200 years) for analysis of some of the models, sometimes including more years of pre-industrial data than mid-Pliocene data for one model. This could impact the results on ENSO variability. We chose to use 100 years of data for all the simulations for the sake of consistency, as some models only had 100 years of data available. The different simulation protocol and ensemble composition makes that differences between PlioMIP1 and PlioMIP2 can be expected. However, the similarities between both ensembles make the reduction of the ENSO amplitude in the mid-Pliocene more robust.

### 4.3  Robustness of ENSO amplitude and significant peaks analysis

Previous studies from Wittenberg (2009) and Tindall et al. (2016) show that ENSO variability can exhibit significant changes on the interdecadal and centennial scale. While a time-series length of 100 years as used here is clearly sufficient to study properties such as amplitude and spectra of ENSO on its typical interannual time scales, the presence of centennial scale modulations on ENSO properties could imply that the choice of the 100-year segment affects our conclusions on interannual variability differences between pre-industrial and mid-Pliocene. In order to address this issue, we repeated the calculation of the ENSO amplitude with 500 years of data. For this we used data of two PlioMIP2 models, MIROC4m and CESM2. The results are included in the Supplement. There are clear variations in the ENSO amplitude on centennial time scales. The Niño3.4 s.d. varies around ±9% and ±4% of the value when using 500 years of data for CESM2 and MIROC4m, respectively. However,



these variations are not large enough to change the conclusion regarding the clear reduction of the ENSO amplitude in the PlioMIP2 ensemble.

Next to this, the spectral 'peak-counting' method is tested for robustness by repeating the procedure using different spectral settings and the Niño3 instead of Niño3.4 index. The conclusions are found to be independent of these different settings. A summary of these results is included in the Supplement. In conclusion, both the reduction of ENSO amplitude as well as the changes in significant spectral peaks in the PlioMIP2 ensemble are robust.

## 4.4    Physical mechanisms

What the PlioMIP2 simulations have in common is the use of identical forcing and (near) identical boundary conditions. The changes in ENSO response and mean state in the mid-Pliocene should therefore be a response to the $CO_2$ forcing and the PRISM4 boundary conditions. Brierley (2015) discusses the impact of the PRISM3 boundary conditions on ENSO and cannot find a clear explanation when considering geography. Differences in boundary conditions from PlioMIP1 to PlioMIP2 mainly include changes to lakes, soils and vegetation, land ice cover and Arctic gateways such as the Bering strait and Canadian

archipelago (Dowsett et al., 2016; Haywood et al., 2016b). We do not expect these differences to directly impact ENSO properties. Other changes to the land-sea mask include the shoaling of the Sahul and Sunda shelves. This can affect the Indonesian throughflow, which can affect ENSO properties as shown by Jochum et al. (2009). We did not investigate the changes in the Indonesian throughflow in the PlioMIP2 ensemble in this study.

       Proxy records suggest the mid-Pliocene mean state sustained 'El Niño-like' conditions (Fedorov et al., 2006; Wara et al.,

2005) or at least a weak zonal SST gradient (Fedorov et al., 2013). Manucharyan and Fedorov (2014) state that the weak zonal SST gradient was less favourable for ENSO occurrence. The PlioMIP2 results do show a reduced zonal SST gradient, but only slightly, together with a reduced ENSO amplitude. However, a consistent relationship between the two is absent, as can be seen in Figure 9. Brown et al. (2020) investigate the relationship between zonal SST gradient and Niño3.4 s.d. for PMIP3/4 simulations of the mid-Holocene, last glacial maximum and last interglacial, as well as 1pct $CO_2$ and abrupt 4x $CO_2$

experiments, and also find no clear relationship. Hu et al. (2013) suggest the relationship between zonal SST gradient and ENSO amplitude might be nonlinear.

       Feedback analysis shows that reduction of ENSO variability in the mid-Holocene is likely due to the increase of the negative feedback (mean current thermal advection, An and Bong (2018)), the reduction of major positive feedback processes (thermocline, zonal advection and Ekman feedbacks, Chen et al. (2019)), or an enhanced seasonal cycle (Iwakiri and Watanabe, 2019).

However, both PlioMIP1 (Brierley, 2015) as well as PMIP3 and PMIP4 models (Brown et al., 2020) do not show any consistent relationship between changes in the seasonal amplitude and changes in ENSO variability. More recently it has been suggested that the mid-Pliocene saw a generally deeper thermocline and thus a weaker thermocline feedback (White and Ravelo, 2020). Brown et al. (2020) acknowledges that the changes in feedback processes in PMIP3/4 simulations are model dependent. They also point out that ENSO feedbacks are still inaccurately simulated in most current GCMs (Bayr et al., 2019; Bellenger et al.,

2014; Kim et al., 2014; Kim and Jin, 2011).





Recent work by Pontes et al. (2021, in prep.), analysing both PlioMIP1 and PlioMIP2 results, find that models with a steeper thermocline (deemed 'La Niña-like' in that paper) are related to major ENSO reduction, while a flatter thermocline ('El Niño-like') is associated with a similar ENSO amplitude compared to the pre-industrial. This finding is in agreement with the results we find on ENSO amplitude and the 'El Niño-like' mean state (Figure 10), although a different methodology is used. Pontes et al. (2021, in prep.) furthermore find that a combination of off-equatorial mean state changes in the mid-Pliocene is not favourable for ENSO. They find a northward displacement of the ITCZ, increased south-eastern trade winds in the west Pacific as well as an intensified South Pacific Subtropical High in the mid-Pliocene simulations, all associated with suppression of ENSO variability.

## 5 Conclusions

The Pliocene Model Intercomparison Project phase 2 (PlioMIP2) coordinated seventeen global coupled climate models to simulate the mid-Pliocene and pre-industrial climate. In this work, we have studied the changes in ENSO variability in the mid-Pliocene using the PlioMIP2 ensemble and related those to SST changes in the tropical Pacific mean climate. Even with (near-) identical forcing and geographical boundaries, the PlioMIP2 ensemble includes a range of model resolutions, parametrizations and initializations, all which contribute to a considerable model spread in the PlioMIP2 ensemble.

The PlioMIP2 ensemble shows a robust decrease in ENSO amplitude. The Niño3.4 s.d. decreases for 15 of the 17 models with an ensemble mean decrease of 24%, compared to 20% in PlioMIP1 (Brierley, 2015). Changes in Niño3.4 skewness and kurtosis are less coherent. Spectral analysis shows a clear reduction of spectral power in the 3 - 4 year period in the mid-Pliocene. Furthermore, the total number of spectral peaks that are above the 99% confidence level decreases with one third in the PlioMIP2 ensemble. The shift towards longer periods that is suggested by the PlioMIP1 models, does not occur in PlioMIP2. The mid-Pliocene ENSO is similar in spatial structure in comparison to the pre-industrial. EOF analysis shows that the percentage of variance explained by the first EOF decreases from 50% to 42% and suggests that El Niño warmth expands meridionally across the tropical Pacific. Correlation with the principal component time series shows that the Niño3.4 index best captures the ENSO-related variability in the Pacific in both the pre-industrial and mid-Pliocene, also suggesting a similar spatial structure. Analysis of El Niño flavour suggests no particular change in the mid-Pliocene, although the model spread is large. Results are very different compared to projections of more central Pacific El Niño events in future climate as suggested by Yeh et al. (2009).

The tropical Pacific mean SSTs show a clear warming in the east Pacific upwelling region, in agreement with proxy data. The PlioMIP2 models do not show a unanimous weakening of the mean zonal SST gradient, with an ensemble mean change of $-0.17 \pm 0.77\,°C$. However, 14 out of the 17 models show a (slight) reduction of the zonal SST gradient. A clear connection with the reduced ENSO amplitude is not found, agreeing with recent work by Brown et al. (2020). Mean SST changes are correlated with the pre-industrial EOF to study if the mid-Pliocene is 'El Niño-like', as suggested by certain proxy studies (Wara et al., 2005; Fedorov et al., 2006). The PlioMIP2 ensemble does not show a unanimous response, and a correlation with the ENSO amplitude and a more or less 'El Niño-like' mean state is not found. Interestingly, there is a 'cluster' of four models (of which



three CMIP6 models) that suggest an 'El Niño-like' mean state but a similar ENSO amplitude in the mid-Pliocene. Proxy data
suggests that the mid-Pliocene ENSO was reduced (Scroxton et al., 2011), similar to present-day (Watanabe et al., 2011), or
possibly varying between both (White and Ravelo, 2020). The PlioMIP2 ensemble shows a range of ENSO amplitudes, varying
between strongly reduced and similar to pre-industrial, possibly even despite a slightly reduced zonal SST gradient. As such,
it seems to generally agree with most proxy studies on ENSO variability as well as tropical Pacific SSTs so far.

Although it becomes more and more clear that the mid-Pliocene saw generally reduced ENSO variability, the reasons for
this are still not clear. White and Ravelo (2020) suggest a weaker thermocline feedback due to a generally deeper thermocline.
Pontes et al. (2021, in prep.) suggest a combination of off-equatorial mean state changes that are unfavourable for ENSO
activity. Future research should aim to find an answer to the question why exactly ENSO variability was reduced in the mid-
Pliocene. These answers can help us to understand the physical behavior of ENSO in warm climates, and will help to understand
behavior of ENSO in the future.

*Code and data availability.* A selection of the data presented in the figures in this paper is available in the Supplement. The 100 years regrid-
ded SST data is available upon request from Alan M. Haywood (a.m.haywood@leeds.ac.uk), as well as more complete data for PlioMIP2
(with exception of IPSLCM6A and GISS2.1G). PlioMIP2 data from CESM2, EC-Earth3.3, NorESM1-F, IPSLCM6A and GISS2.1G can be
obtained through the Earth System Grid Federation (ESGF) (https://esgf-node.llnl.gov/search/cmip6/, last access: 25 February 2021, ESGF,
2021). A selection of the data from the scatter plots presented in the paper are available in the Supplementary material.
Python scripts (Jupyter Notebooks) used for data analysis are freely accessible through github (https://github.com/arthuroldeman/pliomip2-
enso).

*Author contributions.* AMO, MLJB, ASvdH and HAD conceived the presented ideas in this study. AMO performed the analysis and wrote
the draft of the paper. JCT contributed data and advice in data analysis. The remaining authors provided the PlioMIP2 experiments and
contributed to the writing of the paper.

*Competing interests.* The authors declare that they have no conflict of interest

*Acknowledgements.* The work by Arthur M. Oldeman, Anna S. von der Heydt, Michiel L. J. Baatsen and Henk A. Dijkstra was carried out
under the program of the Netherlands Earth System Science Centre (NESSC), financially supported by the Ministry of Education, Culture and
Science (OCW grant no. 024.002.001). Simulations with CCSM4-Utr were performed at the SURFsara Dutch national computing facilities
and were sponsored by NWO-EW (Netherlands Organisation for Scientific Research, Exact Sciences) (project no. 17189).
Alan M. Haywood, Julia C. Tindall, Stephen J. Hunter acknowledge the FP7 Ideas programme: European Research Council (grant no.
PLIO-ESS, 278636), the Past Earth Network (EPSRC grant no. EP/M008.363/1) and the University of Leeds Advanced Research Computing





service. Julia C. Tindall was also supported through the Centre for Environmental Modelling and Computation (CEMAC), University of Leeds.

Bette L. Otto-Bliesner, Esther C. Brady and Ran Feng acknowledge that material for their participation is based upon work supported by
the National Center for Atmospheric Research, which is a major facility sponsored by the National Science Foundation (NSF) (cooperative agreement no. 1852977 and NSF OPP grant no. 1418411). Ran Feng is also supported by NSF grant no. 1903650. The CESM project is supported primarily by the National Science Foundation. Computing and data storage resources, including the Cheyenne supercomputer (https://doi.org/10.5065/D6RX99HX), were provided by the Computational and Information Systems Laboratory (CISL) at NCAR. NCAR is sponsored by the National Science Foundation.

Ning Tan, Camille Contoux and Gilles Ramstein were granted access to the HPC resources of TGCC under the allocations 2016-A0030107732, 2017-R0040110492 and 2018-R0040110492 (gencmip6) and 2019-A0050102212 (gen2212) provided by GENCI. The IPSL-CM6 team of the IPSL Climate Modelling Centre (https://cmc.ipsl.fr, last access: 28 April 2021) is acknowledged for having developed, tested, evaluated and tuned the IPSL climate model, as well as per- formed and published the CMIP6 experiments.

Christian Stepanek acknowledges funding from the Helmholtz Climate Initiative REKLIM. Christian Stepanek and Gerrit Lohmann
acknowledge funding via the Alfred Wegener Institute's research programme Marine, Coastal and Polar Systems.

Qiong Zhang acknowledge support from the Swedish Research Council (2013-06476 and 2017-04232). Simulations with EC-Earth were performed on resources provided by the Swedish National Infrastructure for Computing (SNIC) at the National Supercomputer Centre (NSC).

Wing-Le Chan and Ayako Abe-Ouchi acknowledge funding from JSPS (KAKENHI grant no. 17H06104 and MEXT KAKENHI grant no. 17H06323). Their simulations with MIROC4m were performed on the Earth Simulator at JAMSTEC, Yokohama, Japan.

W. Richard Peltier and Deepak Chandan wish to acknowledge that data they have contributed from the CCSM4-UoT model was produced with the support of Canadian NSERC Discovery Grant A9627 t WRP, and they wish to acknowledge the support of SciNet HPC Consortium for providing computing facilities. SciNet is funded by the Canada Foundation for Innovation under the auspices of Compute Canada, the Government of Ontario, the Ontario Research Fund – Research Excellence, and the University of Toronto.

Zhongshi Zhang and Xiangyu Li acknowledge financial support from the, National Natural Science Foundation of China (grant no.
42005042), the China Scholarship Council (201804910023) and the China Postdoctoral Science Foundation (project no. 2015M581154). The NorESM simulations benefitted from resources provided by UNINETT Sigma2 – the National Infrastructure for High Performance Computing and Data Storage in Norway.

Charles J.R. Williams and Dan Lunt thank NERC grant NE/P01903X/1, and the NEXCS High Performance Computing facility funded by the Natural Environment Research Council and delivered by the Met Office.
Gabriel M. Pontes and Ilana Wainer acknowledge the São Paulo Research Foundation (FAPESP 2016/23670-0).



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
