# Peer review of "Reduced El Niño variability in the mid-Pliocene according to the PlioMIP2 ensemble"

_Climate of the Past, 2021_

## Author Response (AR2)

**Author's response 2**

September 21st 2021

Dear editor,

Many thanks for your comments regarding the revised version of our manuscript. Please find the responses to your comments below.

Also, the previous Author's response as well as point-by-point responses to the reviews (as published on CP Discussions on behalf of all co-authors) are included on the following pages.

Attached you will also find the revised manuscript as well as a 'track changes' version (only including changes with the previously revised version).

I hope you will accept the manuscript in the present form. Thank you for considering publishing our work in Climate of the Past.

Best regards, on behalf of all the co-authors,

Arthur Oldeman

**Response to editor comments**

Dear authors,

Thank you for submitting a revised version of your manuscript.

I have reviewed the revised manuscript, including your responses to the comments by the referees, and I consider that it is now very close to the point where I can accept it for final publication in Climate of the Past.

However, before I do so, please consider the following comments on your responses to Referee #1 (Chris Brierley). In the following, I use the same line and section numbers that the referee uses in his review.

S3.1.2, L358, S3.2.3 and L605: In each of these four cases, you include useful information in your responses but have not incorporated it into the manuscript. Could you consider adding at least some of this information?

- **S3.1.2 (lengthening of spectral periods): Regarding this response, we consider the period of maximum spectral power to not be so relevant (and explain why in the response). However, we understand that this reasoning might be useful to the reader. We therefore include a concise version of the response in the revised manuscript at L273-L278:** *"We find that 12 out of the 17 models show an increase in the period of maximum spectral power in the mid-Pliocene, compared to the pre-industrial. However, ENSO does not have one isolated period in the power spectrum. While there are models that show peaks in spectral power in the mid-Pliocene at higher periods that in their pre-industrial counterparts, in many cases these spectral peaks do not exceed the threshold for statistical significance. Therefore, we only consider the ensemble mean power spectrum as well as the histogram of significant spectral peaks here."*

- **L358 (reference to choose cold tongue / warm pool regions): After considering this response again, we will slightly reformulate to prevent confusion about the cold tongue and warm pool regions being 'known' or 'set' regions. We will change the formulation from** *'western Pacific warm pool (…)'* **to** *'the 'warm pool' region in the equatorial western Pacific Ocean (…)'* **and similar for the cold tongue region. We made changes in the revised manuscript in the Methods (L222-L223) and in the Results (L390-L392).**

- **L3.2.3 (lines of best fit in Figure 10): We do not think that the information in this response needs to be added to the manuscript. The relation between ENSO amplitude and zonal SST gradient is shown in Fig9b and already treated in the text. The relation between ENSO amplitude and 'El Niño-like' mean state changes is treated more in Fig11a-d as well as in the accompanying test.**

- **L605 (24% reduction distinguishable from 20% reduction?): Regarding this response, we consider the standard deviation on the ensemble mean ENSO amplitude reduction to be relevant to add to the text and have added it in the Results section 247):** *"The mid-Pliocene ensemble mean shows a 24% reduction in ENSO amplitude (compared to 20% in PlioMIP1, (Brierley (2015)) with a standard deviation of ±18%."* **As well as a mention in the Discussion in L577**

L123: Some of the information in your response has been added to the manuscript, but I think it would be helpful to include more. In particular, I think the information "… the majority of the models use an ocean resolution that is close to 1°x1°. Only IPSLCM6A-LR uses a 1/3° ocean resolution in the tropics, where the smoothing out of the extremes could play a role." would be useful for the reader.

**Thank you, and excuse us for not including this relevant information. It is because we realised only later that not only IPSLCM6A-LR has a finer horizontal ocean resolution in the tropics, but also some other models employ a telescoping grid that refines near the equator (in the meridional direction). We decided to remove that part and were planning to include a more detailed explanation. We will now include a more detailed explanation in the revised manuscript (L141-L144):** *"The majority of the models use a nominal horizontal ocean resolution that is close to 1°. Some of the models employ a telescoping grid that results in a finer resolution near the equator, such as IPSLCM6A-LR as well as CCSM4-Utr (~ 0.33° and ~ 0.67°, respectively). Since we are not interested in gridbox-scale features, and often considering spatial means as well as ensemble means, we do not expect the regridding to significantly impact the results."*

L165: "normalisation" is indeed the usual convention in climate studies. I am happy for you to retain the use of the word "standardisation", but it would be helpful to expand the sentence in the manuscript to clarify that the standardised indices have zero mean and unit variance.

**Thank you for the feedback. We changed the formulation to 'normalisation' and added a brief clarification in L185-186 of the revised manuscript:** *"Before the power spectra are computed, the indices are normalised with their s.d. to obtain time series with zero mean and unit s.d."*

L274: In your response, you state "Next to this, the EOF patterns are scaled with their standard deviation, therefore removing the ENSO amplitude. This is done in order to compare the spatial pattern only.". I don't see this information in the manuscript (apologies if I have missed it). This would be useful information for the reader, so please add it.

**Thank you, this was already very briefly mentioned in the Methods section ('… performing EOF analysis and normalisation …'). We decided to include a clarification in this section, instead of in the Results. This sentence (L205-L206 in the revised manuscript) now reads:** *"We follow the methodology of Power et al. (2013) in performing EOF analysis and normalising the EOF with the spatial s.d. in order to remove the ENSO amplitude signal, as did Brierley (2015) for the PlioMIP1 ensemble."*

L340: I also find the use of the letters P-U, without explanation, to be extremely confusing. I don't personally consider that it would be confusing to use A-F instead. However, if you are concerned about potential confusion with subfigures, could you consider a solution such as "P1" to "P6"?

**Thank you for the feedback, we have decided to use A-F instead of P-Q and made some clarifications in the caption of Figure 7 as well as in the text.**

L545: In the revised text, the use of the word "confidence" by itself is a little vague. Could you consider alternatives, such as "statistical confidence", "confidence intervals" or "degree of confidence"?

**Thank you, this is a fair point. We made sure that the word 'confidence' is not used without context. In practice, this means that only in L588 of the revised manuscript we added 'statistical' before 'confidence'.**

Kind regards,
Steven Phipps
Handling Editor

**Author's response 1**

August 29th 2021

Dear editor,

Please find the point-by-point to the reviews (as published on CP Discussions on behalf of all co-authors) on the following pages.

Attached you will also find the revised manuscript as well as a 'track changes' version.

I will list here the major changes made in the revised version:
- Incorporation of recent ENSO and tropical Pacific SST variability research with CMIP6 data (in Introduction, as well as Results and Discussion)
- Clarification of earliest modelling studies on Pliocene ENSO variability (Introduction 1)
- Addition of a comment on interpolating on a common grid (Methods 2.1)
- Clarification on insufficient model resolution to reproduce coastal upwelling systems (Results 3.2.1, as well as in the Discussion 4.1.2)
- Addition of a comment on possible overlap of Nino3.4 s.d. change results between models contributing to both PlioMIP1 and PlioMIP2, using the 500 year robustness analysis (Discussion 4.3)
- Addition of references on the influence of the Indonesian throughflow and Bering strait on ENSO properties (Discussion 4.4)
- Revised all figures: increased visibility (e.g. increased fontsize, colormaps, linestyles)

I hope you will accept the manuscript in the present form.

Thank you for considering publishing our work in Climate of the Past.

Best regards, on behalf of all the co-authors,

Arthur Oldeman

**Response to Referee Comment 1 (Chris Brierley)**

**Dear Chris Brierley,**

**Thank you for your constructive comments and feedback. We will answer your specific questions and comments here and indicate how we plan to make any changes in the revised manuscript.**

My main issue with the research as written is that it does not incorporate the recent literature on the subject of modelling ENSO. Recent advances relating to the paleoclimate component of CMIP6 have been included, but there is a raft of publications looking at ENSO in the historical and future scenarios that are not considered. This is particularly noticeable with the discussion of Yeh et al (2009) - there are certainly more recent works looking at future changes in ENSO flavour. These are reviewed in the upcoming IPCC AR6 and I strongly advise waiting until that is published before completing your revisions. **– This is a fair point, and we will include a number of references to more recent publications on ENSO in the historical and future scenarios, such as Frederiksen et al. (2020), Freund et al. (2020), Beobide-Arsagua et al. (2021) and Jiang et al. (2021):**
**https://doi.org/10.1029/2020GL090640**
**https://doi.org/10.1175/JCLI-D-19-0890.1**
**https://doi.org/10.1007/s00382-021-05673-4**
**https://doi.org/10.1175/JCLI-D-20-0551.1**
**If available when preparing the revised manuscript, we will also consider the IPCC-AR6 report of WGI.**

Specific comments:

L22. You give the date for the whole mPWP here, but never explicitly mention that PlioMIP is aimed at an interglacial within this period. KM5c is mentioned in passing later on, but without dates. **– We will include the specific years of the KM5c time slice in the Introduction and explicitly mention why this time slice was chosen.**

L37. You need to described what the bracket with "(1.7-5.2oC)" indicates. Does it include HadGEM3-GC31-LL? **– It shows the min and max values of the global mean SAT difference within the ensemble. This is not including the HadGEM3 contribution, we will clarify the numbers and use the updated values (including HadGEM3) from Williams et al. (2021):**
**https://cp.copernicus.org/preprints/cp-2021-40**

L60. Did either of Fedorov et al and Barriero et al actually use coupled models? If not, how could be expected to any variability? **- Fedorov et al 2006 is a review paper, citing model studies (mainly not-coupled) to show that a permanent or perennial El Niño state could have been possible in the mid-Pliocene. Barreiro et al used a forced atmosphere-only GCM. You are correct in noting that these studies do not resolve ENSO variability, and we will rephrase that in the manuscript.**

L68. (or longer periods) -> (so longer periods) **– Will be corrected**

L85. Brown et al (2020) did not look at future scenarios. Rather they used the idealised warming scenarios of the CMIP DECK. **– True, will be corrected**

L123. Interpolating variables onto a common grid prior to analysis is not best practice. This would act to smooth out spatial variations and lop-off extremes. I do not expect you to re-perform all of your analysis, as I suspect that it will make little difference to your conclusions. You may want to mention why this is the case in your methods section. Try to avoid this approach in future – it should be performed at the last possible moment, as part of the ensemble averaging. **– Thanks for mentioning that. We do not expect this to affect the results greatly, as the majority of the models use an ocean resolution that is close to 1°x1°. Only IPSLCM6A-LR uses a 1/3° ocean resolution in the tropics, where the smoothing out of the extremes could play a role. We already clearly state in the methods section that we use the regridded data, and we will add some discussion of why we expect this to not impact our conclusions.**

Table 1. Why are you not using the model acronyms that are part of the CMIP controlled vocabulary? Will this not prevent your study coming up on Google Scholar searches and the like? **– It is chosen to follow the PlioMIP1/2 naming conventions and to be consistent with other PlioMIP2 studies. However, we will include CMIP vocabulary in the Table (when it is different from the PlioMIP naming).**

Table 1. Are you sure that HadCM3 was a CMIP5 model? I thought it was CMIP3. **– HadCM3 can be considered a CMIP3 generation model, but it did contribute to CMIP5.**

L155. Factor 3.0 -> factor of 3.0 **– Will be corrected**

L165. I believe that "standardised" should be "normalised" here. **– Dividing the Nino3.4 indices by their standard deviation causes them to have a zero mean (anomalies) and s.d. of 1. In statistical studies, this is actually called standardisation and not normalisation, but there could be a different naming convention in climate studies.**

L162. How does your statement about the internal variation compare with the conclusions of Tindall et al? **- The conclusions are essentially the same. In the Discussion (L556-559) we state: "There are clear variations in the ENSO amplitude on centennial time scales. […] However, these variations are not large enough to change the conclusion regarding the clear reduction of the ENSO amplitude in the PlioMIP2 ensemble." Tindall et al state (in the abstract): "The Pliocene-preindustrial El Niño temperature […] and precipitation signal are usually larger than centennial-scale variations of El Niño amplitude and provide consistent indications of ENSO amplitude change."**

L182. Please mention if the monthly SST anomalies are detrended prior to the PCA. **– We did not detrend the SST anomalies prior to the PCA (although it is stated we do in L185). We redid the PCA with a linear trend removed from the SST anomalies and find no significant changes (the error is in the order of 1e-3 on EOF of order 1). The difference in the percentage of variance explained is in the order of 0.1%. We will include the new (detrended) EOFs in the figures of the revised manuscript.**

L230. GISS-E2.1-G was not in PlioMIP1 – rather that was GISS-E2-R. Please justify why you consider these to be iterations of the same model, rather than different generations as other studies often do.  - **Thank you for the clarification, we will remove the mention in L234**

Sect 3.1.2 How many of the 17 model show an lengthening of the periods – refer to Fig. S2 for this. **– If we consider the period of maximum spectral power as the ENSO period, we find that 12 out of the 17 models show an increase in the period in the mid-Pliocene. However, ENSO does not have one isolated period in the power spectrum, as we show in Fig. S2. While there are models that appear to show peaks in spectral power in the mid-Pliocene simulations at lower frequencies than in their pre-industrial counter parts (EC-Earth3.3, GISS2.1G, HadCM3, IPSLCM5A, IPSLCM6A, MIROC4m, and MRI2.3), in many cases these peaks do not exceed the threshold for statistical significance. For example, IPSLCM5A has their maximum Eoi400 peak at 9 years, but this specific peak does not exceed the 90%-CI. Consequently, it is difficult to provide robust conclusions on the significant spectral changes per ensemble member. We therefore prefer to stick to the methodology of binning in the 1.5-10 year period range (Fig. 4), focussing on those peaks that are indeed significant according to our analysis. This peak counting procedure is not particularly meaningful when performed for one model, as the number of significant peaks can be low (as low as 1 peak above the 99%-CI for EC-Earth3.3's Eoi400 spectrum), thus making it difficult to provide robust conclusions as well.**

L274. Please provide more explanation about the word "normalised" - is the information about the ENSO amplitude (in oC) contained within the EOF or the PC? **– Here it means that EOFs are scaled to be positive in the Nino3.4 region. We will clarify this. Next to this, the EOF patterns are scaled with their standard deviation, therefore removing the ENSO amplitude. This is done in order to compare the spatial pattern only.**

L282. Cite Fig. S3 to support this. **– OK, we will do this.**

L289. Please rephrase to only use word "region" to have a single meaning. **– We will use 'area' here instead.**

L309. Please compare this HadCM3 result with that shown in Brierley (2015). **– HadCM3 shows similar results in PlioMIP1; similar warm pool and cold tongue s.d. in the pre-industrial, and relative increase in warm pool s.d. in the mid-Pliocene.**

L325. This sentence reads as if it encompasses the warmer E. Pac coastal temperatures. These are instead a feature of insufficient ocean model resolution to capture the coastal upwelling.  **– We will correct this: 'along the east Pacific coast' → 'in the east Pacific'. Furthermore, we will rephrase the next sentence: 'The 'cold bias' in the east Pacific can be expected since, firstly, the pre-industrial simulations are compared with historical observations and, secondly, the models have insufficient resolution to reproduce the cold conditions of coastal upwelling systems, such as the Benguela upwelling system (McClymont et al. 2020)' McClymont et al provides an overview of the known model deficiencies with regards to the coastal upwelling. We will add a sentence in the**

**Discussion on the model and observation mismatch in upwelling regions and the ocean resolution needed to resolve this better, referring to Small et al. (2014): https://doi.org/10.1002/2014MS000363**

L333. Choosing a red-green color scheme is unhelpful to readers who are colorblind. **– We will make sure to correct this.**

L340. The alphabetic indictors have not been introduced earlier. Why do you start at P? Please add letters to Fig 7. **– The alphabetic indicators or letters are included in the circles Figure7d. We will enlarge them slightly for increased visibility. We started at 'P' instead of 'A' in order to avoid confusion with the subfigure count.**

L352. Warming trends (up to average year of 1970) are less than 1oC globally, let along tropical pacific. Put nuance in your expectation. **– We will compute the average equatorial Pacific SST difference between the pre-industrial simulations and the HadISST observations instead of providing an estimate now. Do note that the HadISST data range we have chosen to include (1920-2020) does not cover the full pre-industrial period and may thus show relative larger warming than when including the full historical period.**

L358. Please be consistent with your longitude names. Fig 7, showing these boxes, goes 0--360 not -180--180. **– We will make sure to be consistent in the full manuscript.**

L358. Is there a reference to choose these regions? You later discuss how these regions are inappropriate for 2 models. Maybe using max and min in two larger region would be more helpful? **– We have chosen the two regions based on the ensemble mean and HadISST equatorial SSTs as shown in Figure 8, such that we expect most models to have their minimum and maximum SSTs in one of the regions. We did not choose these regions based on a reference. Actually, only for MRI2.3 one can say that the regions are a poor choice. But the reason here is that MRI2.3's equatorial Pacific SSTs in the pre-industrial are an outlier, when comparing to the ensemble mean as well as the HadISST result. We could use a larger region indeed, but this would also cause a smoothing out of the min and max values.**

Fig 8. prints badly in black & white, I've discovered. Also would be poor color choice for those suffering from deuteranomaly. **– We will change linestyle in the figure to improve distinction between lines and possibly add hatching or stippling in the shaded area (if the figure does not get too crowded).**

L375. I feel that it is worth stressing that Brown et al include many of the models used here. **– Agreed, will include this.**

Sect 3.2.3 Any lines of best fit would not pass through the origin in either Fig 10a or 10b. What are the implications for that on your interpretation? Are you expecting an external condition to cause a roughly 25% amplitude reduction and then the zonal gradient to control the deviations from that? **– To clarify: Fig10a and b show the ENSO amplitude change as a function of the mean state changes to pre-industrial EOF correlation, not as a function of the zonal SST gradient. The relation between ENSO amplitude and zonal SST**

gradient is show in Fig9b and is treated in the text (L373-375). In the case of Fig 10a and 10b, a line of best fit not passing through the origin would imply that an ENSO reduction is possible even when the changes in annual mean SST are uncorrelated to the pre-industrial EOF pattern. When considering the results of both Fig 10a and 10b together, it seems that ENSO reduced in the mid-Pliocene simulations and that this reduction does not seem to be correlated to the mean state being more or less 'El Niño-like'.

L394. Pre-industrial -> E280? **– We will change this for clarification**

L399. Fig 11d does not show 'reduction' in ENSO amplitude, as that implicitly is between E280 and Eoi400. Please rephrase sentence. **– Agreed, this is shown in Fig10a but not in Fig11d, we will change the formulation.**

Sect 4.1.1 You do not mention the observational uncertainty in the earliest portion of the record (e.g. Ilyas et al, 2017: https://agupubs.onlinelibrary.wiley.com/doi/full/10.1002/2017GL074596). I wonder if we really can resolve ENSO flavour accurately back in the 1920s. **– We will refer to this work in the discussion. In fact, it provides a good argument for using the most recent data (1920-2020) instead of a period that would represent the pre-industrial era.**

L440. Is "clear agreement" the best choice of words. I know what you mean – but isn't more like "cannot rule out disagreement" **– We will formulate it as: "Taking into account these uncertainties, the model differences fall within the range of the reported error around the observational differences (proxy minus HadISST) as reported in Table 3, …"**

L447. This sentence needs a citation. **–The statement on discrepancies with proxies in mid- and higher latitudes is supported by the two citations (Haywood et al. 2020, McClymont et al. 2020) in the following sentence:**
**https://doi.org/10.5194/cp-16-2095-2020**
**https://doi.org/10.5194/cp-16-1599-2020**

L450-2. I do not see how both these sentence can be true. Isn't a reduction in SST gradient the defining feature of an El Nino-like change? Please reconcile these facts. - **The zonal SST gradient is not the only feature defining an 'El Nino like' change. To quantify this better, we have chosen to investigate if the mean SST changes are 'El Nino-like' by correlating them with the E280 leading EOF. What we refer to in line 450-452 is that 1) the ensemble shows a slight reduction of the zonal SST gradient (based on the results in Fig9) and that 2) the mean state changes are not specifically 'El Nino-like', as the majority does not correlate positively with the E280 leading EOF. We will reformulate these two sentences so that this is clear.**

L456. Personally, I feel that Watanabe et al.'s conclusions on this point overstepped their data. **– We use the phrasing as used in their paper and say that it is 'suggested'**

L505. Difference -> different **– Will be corrected (line 508)**

L545. Spell out how the longer, yet inconstant, record lengths I used in 2015 could impact the results. Also, are they detrimental impacts? **– We will rewrite the sentence: "This could impact the results on ENSO variability" → "This can impact the confidence of spectral power density, particularly for the longer periods." Longer time series could reveal more confident lower frequency power density. When using inconsistent time series length (especially between pre-industrial and mid-Pliocene of one model), this can affect the spectral comparison. It could be detrimental when considering the longer periods (>10y on a time series length of 100y), but we do not consider those in the manuscript.**

L560. The analysis of the 500 year long records is a really nice addition to the paper. I was left wondering what the impact of internal variability could be on Fig. 10. You could readily assess this, and that might explain why the overlapping models in a & b differ (although note GISS is not overlapping). **– Thanks and yes, that might be a nice thing to add; when considering the +-9% variation on the s.d., MRI2.3, MIROC4m and IPSLCM5A show overlap in range with their PlioMIP1 values, but HadCM3, CCSM4 and NorESM-L do not. Indeed – GISS is a different model version, we will change the marker used in Fig10a and b for the two GISS contributions.**

L572. The Indonesia throughflow impacts were also investigated by myself (Brierley & Fedorov, 2016) and Zhongshi Zhang (certainly Zhang et al 2016, but maybe others). **– Thanks for mentioning, we will include a slightly more elaborate discussion on the topic and include more references, such as Brierley & Fedorov (2016) and Karas et al. (2009): https://doi.org/10.1016/j.epsl.2016.03.010 https://doi.org/10.1038/ngeo520**

L605. Is your 24% reduction distinguishable from my 20% reduction? Can you use your analysis of the 500 year simulations to estimate a significance? **– Considering the range of % change in Nino3.4 s.d. as shown by the different ensemble members (resulting in an ensemble mean change of -24% with a standard deviation of +-18%), we would say that no, this is not distinguishable. We do not think the 500y analysis can be used for this purpose as this only gives a result for 2 of the 17 PlioMIP2 models. It could be used for an estimate, but only for the PlioMIP2 ensemble, not for the PlioMIP1 ensemble as there are also differences in boundary conditions between the two ensembles.**

**Response to Referee Comment 2 (anonymous referee)**

**Dear anonymous referee,**

**Thank you for your nice remarks and your feedback. We will answer your specific questions and comments here and indicate how we plan to make any changes in the revised manuscript.**

This paper provides a comprehensive discussion of ENSO variability in the new PlioMIP2 ensemble. The changes in ENSO amplitude, frequency and spatial pattern and the relationship with mean state changes are all evaluated and compared with proxy records and other modelling studies. The results will be of wide interest. The paper is fluently written and well structured. The model evaluation uses a thorough and logical methodology in all sections. I recommend the paper for publication subject to minor revisions outlined below.

Specific comments:

Model names: The model names in Table 1 are not the same as the names in Table 2 and throughout the paper. I suggest using the standard CMIP5/CMIP6 names throughout to avoid confusion. **– Similar comment as RC1, my response here: It is chosen to follow the PlioMIP1/2 naming conventions and to be consistent with other PlioMIP2 studies. However, we will include CMIP vocabulary in the Table (if it is different from the PlioMIP naming).**

Introduction, line 24 onwards: The discussion of mid-Pliocene $CO_2$ levels is a bit misleading as the de la Vega study found that "$CO_2$ranged from 394 (+34/-9) ppm to 330 (+14/-21) ppm: with $CO_2$ during the KM5c interglacial being 391 (+30/-28) ppm (at 95% confidence)." Therefore, for the PlioMIP2 interval, the new estimates overlap with the value of 400 ppm given in the earlier studies. Haywood et al. (2016) could also be cited here for providing a discussion of the range of $CO_2$ estimates – their discussion also concludes that the various $CO_2$ estimates generally overlap. **– Agreed that this passage can be improved, we will rephrase it with your recommendations.**

Section 2.1: It should be mentioned here that the PlioMIP2 Eoi400 simulations are configured to represent the KM5c interglacial at 205 Ma, and that this interval has an orbital configuration close to the present day allowing models to use the same orbital parameters as for the pre-industrial simulation. **– Agreed, similar comment as RC1, we will make sure to include the configuration in the Introduction, when mentioning the PlioMIP2 ensemble.**

Technical corrections:

Line 37: change "increases with" to "increases by". **– Will be corrected**
Line 67: change "accompanied with" to "accompanied by". **- Will be corrected**
Line 106: change "will be" to "are" and delete "will" before "conclude" for consistency with tense in the rest of this paragraph. **- Will be corrected**
Line 217: Suggest change subsection 3.1.1 title to "Statistical moments" rather than "Moments". **– We will take over this suggestion**

Line 371-372: Suggest summarize the comment on the MRI2.3 zonal SST gradient here, e.g. "due to the overly westward extent of the cold tongue in the MRI2.3 model discussed above".- **Agreed, that might read better, will reformulate the sentence.**

Line 397: Change "alike" to "like". - **Will be corrected**

Line 461: Insert "when comparing mid-Pliocene and pre-industrial ENSO" after "robust signals". - **Will be corrected**

Line 512: Change "great similarity for" with "great similarity between". – **Will be corrected**

Line 547: change "composition makes that" with "composition means that".- **Will be corrected**

Figures: Axis labels are very small, consider increasing font size. – **Agreed, we will improve readability of the figures.**

Figures: The use of red-green color scheme may not be suitable for colorblind readers. Perhaps use red-blue instead. – **Similar comment as RC1, we will choose another colormap.**

Figure 1: Make Nino box lines thicker as they are hard to see. - **Will be corrected**

References: Check reference formatting as some information is duplicated. – **It seems that for some references the doi is duplicated, we will correct this.**